# Zebrafish Avatar-test forecasts clinical response to chemotherapy in patients with colorectal cancer

Bruna Costa [1], Marta F. Estrada [1], António Gomes[2,9], Laura M. Fernandez[3,9], José M. Azevedo [3], Vanda Póvoa [1], Márcia Fontes [1], António Alves [4], António Galzerano[5], Mireia Castillo-Martin[5], Ignacio Herrando [3], Shermann Brandão[6], Carla Carneiro[2], Vítor Nunes[2], Carlos Carvalho[6], Amjad Parvaiz[3], Ana Marreiros [7,8] & Rita Fior [1] ✉

Cancer patients often undergo rounds of trial-and-error to find the most effective treatment because there is no test in the clinical practice for predicting therapy response. Here, we conduct a clinical study to validate the zebrafish patient-derived xenograft model (zAvatar) as a fast predictive platform for personalized treatment in colorectal cancer. zAvatars are generated with patient tumor cells, treated exactly with the same therapy as their corresponding patient and analyzed at single-cell resolution. By individually comparing the clinical responses of 55 patients with their zAvatar-test, we develop a decision tree model integrating tumor stage, zAvatar-apoptosis, and zAvatar-metastatic potential. This model accurately forecasts patient progression with 91% accuracy. Importantly, patients with a sensitive zAvatar-test exhibit longer progression-free survival compared to those with a resistant test. We propose the zAvatar-test as a rapid approach to guide clinical decisions, optimizing treatment options and improving the survival of cancer patients.

Colorectal cancer (CRC) is the third most common cancer and a leading cause of cancer-related deaths worldwide[1]. Although most surgeries have a curative intent, circulating tumor cells or undetectable micrometastases can be present after surgery. Thus, patients with high-risk factors for metastatic disease have been shown to greatly benefit from post-surgical systemic therapies to reduce the likelihood of relapse and disease progression[2]. A combination of 5-fluorouracil (5-FU) with folinic acid and either irinotecan (FOLFIRI) or oxaliplatin (FOLFOX) is the standard systemic chemotherapy for advanced or metastatic CRC (mCRC). These regimens are generally considered interchangeable and variations of these combinations exist when introducing orally active FU-like drugs, such as capecitabine: CAPOX (capecitabine+oxaliplatin), and CAPIRI (capecitabine+irinotecan). In the last 15 years, monoclonal antibodies, such as cetuximab or bevacizumab, have also been included in first-line chemotherapy regimens[3–5].

Randomized clinical trials have shown that FOLFOX and FOLFIRI are equivalent options for advanced CRC treatment, with similar average response rates (RR) ranging from 34 to 55%[3,6–8]. This means that approximately 45 to 66% of patients do not respond to treatment.

[1]Champalimaud Research, Champalimaud Foundation, Lisbon, Portugal. [2]Surgery Unit, Hospital Prof. Doutor Fernando Fonseca, Amadora, Portugal. [3]Colorectal Surgery Department, Champalimaud Clinical Centre, Champalimaud Foundation, Lisbon, Portugal. [4]Institute of Pathological Anatomy, Faculty of Medicine of the University of Lisbon, Lisbon, Portugal. [5]Pathology Service, Champalimaud Clinical Centre, Champalimaud Foundation, Lisbon, Portugal. [6]Digestive Unit, Champalimaud Clinical Centre, Champalimaud Foundation, Lisbon, Portugal. [7]Faculty of Medicine and Biomedical Sciences, University of Algarve, Faro, Portugal. [8]Algarve Biomedical Center Research Institute, University of Algarve, Faro, Portugal. [9]These authors contributed equally: António Gomes, Laura M. Fernandez. ✉e-mail: rita.fior@research.fchampalimaud.org

For instance, if patients start with FOLFOX as first-line and do not respond to treatment, they can switch to FOLFIRI, and vice versa. Consequently, many patients suffer unnecessary side effects and lose valuable time.

This issue extends beyond CRC to many other types of cancers, where several "equivalent" treatment options are available in the guidelines but lack a reliable predictive test to forecast the outcome and aid the clinical-decision process. With exception of some success cases, current cancer molecular and genetic biomarkers have proven insufficient when it comes to reliably predicting treatment outcomes. It has been shown that even genetically identical CRC cells may have differential response to therapy, implying that the basis for therapy response is not only genetic[9]. Many cancer patients do not benefit from genomic precision medicine due to a combination of factors, including the absence of targetable mutations, the lack of effective drugs for specific promising targets and also the possible genetic interactions that may occur between different tumor subclones or with the tumor microenvironment[10,11]. Thus, a combination of molecular-profiling precision medicine together with a functional test, where tumor cells are directly challenged with the elected therapies, is fundamental for a more accurate personalized medicine[10,12].

We have been developing a fast in vivo functional test with unprecedented cellular resolution—the zebrafish patient-derived xenograft model or zAvatar[13]. This assay relies on the injection of fluorescently labeled patient tumor cells into 2 days post fertilization (dpf) zebrafish embryos. Among its numerous advantages, the most important are the ability to analyze metastatic and angiogenic potentials in vivo, and the speed of the test: tumor behavior and response to therapy can be accessed in just 10 days, a time frame compatible with oncological clinical decisions[14–17]. Previous studies, including our own, have shown that zAvatars can predict patient clinical outcomes[13,18–25]. Although promising, these results were obtained from a limited number of patients. Thus, larger co-clinical studies are fundamental to validate the use of zAvatars for predicting individual treatment response.

Here, we present the results of a co-clinical study where chemotherapy clinical response of 55 CRC patients is individually compared with the corresponding zAvatar-test. Our data demonstrate that the zAvatar-test successfully predicts the outcome of 50 out of 55 patients, anticipating either no-progression/stable disease or disease progression after systemic therapy. Multivariate analysis reveals that three specific parameters are the most important predictors of patient clinical response: patient's tumor stage, zAvatar metastatic potential and zAvatar apoptosis fold change. By integrating only these variables into a decision tree algorithm, the zAvatar-test achieves a positive predictive value (PPV) of 91% and a negative predictive value (NPP) of 90%, demonstrating the fidelity of the model in mirroring patient outcomes. Thus, we propose the zAvatar-test as a fast tool to guide clinical decisions, optimizing the current standard of care and improving progression-free survival of many cancer patients.

## Results

### Clinical study design, patient cohort, and examples of zAvatars analysis

To assess the predictive power of the zAvatar model, we conducted a co-clinical study where patient's chemotherapy clinical response was individually compared with their corresponding zAvatar-test (Fig. 1a). A total of 79 patients diagnosed with CRC who underwent systemic chemotherapy after surgery were enrolled. We were able to successfully perform the zAvatar-test in 55 patients, which corresponds to a 70% of technical success rate. The main reasons for nonsuccess were a small initial tumor sample, sample necrosis or death of zAvatars during the experiment. Patients whose zAvatars had low implantation ($n < 4$

zAvatars with tumors for each condition) were excluded from the study. Patient samples include 50.9% colon cancers ($N = 28$), 12.7% rectal cancers ($N = 7$), and 36.4% liver metastases ($N = 20$), including 9.1% stage II ($N = 5$), 49.1% stage III ($N = 27$) and 41.8% stage IV ($N = 23$) (Fig. 1b, c and Supplementary Table 1). The recruited patient cohort shows a balanced representativeness in terms of stages, type and subtype of tumors, mutations, and other characteristics (Supplementary Table 1 and Supplementary Data 1).

All patients were treated according to standard of care, with adjuvant or perioperative chemotherapy to reduce the chances of relapse and disease progression, based on NCCN/ESMO clinical guidelines decided in a multidisciplinary team meeting.

Zebrafish Avatars were generated with patient tumor cells and treated with exactly the same therapy as their corresponding patient. At 3 days post-injection (dpi) and 2 days post treatment, several readouts were analyzed and compared between untreated-controls and treated zAvatars such as: induction of apoptosis (activated caspase 3), tumor size fold change, formation of micrometastases, and tumor implantation/persistence (Supplementary Fig. 1). Labeling human cancer cells with lipophilic dies can be very heterogenous, with the brightest cells often being apoptotic, and consequently susceptible to uptake by phagocytes (Supplementary Fig. 2). Thus, to ensure specificity in identifying the human patient cells we use the anti-human mitochondria antibody as a quality control. zAvatar response to treatment was blindly compared with patient clinical response 12 months after starting chemotherapy (Fig. 1a- see "Methods" section).

In Fig. 1, we present two examples of patients with different treatment outcomes and their corresponding zAvatar-test. Patient #138CCU with a right colon adenocarcinoma was treated postoperatively with CAPOX. The zAvatar-test showed an average of 2-fold induction of apoptosis upon CAPOX treatment, in relation to untreated zAvatar controls. This patient showed no progression i.e., had no signs of distant disease or local recurrence (Fig. 1e–h and Supplementary Fig. 3). In contrast, zAvatars from patient #239AS, diagnosed with a right colon adenocarcinoma and liver metastasis, showed no response to FOLFOX treatment, since it did not induce tumor apoptosis. Accordingly, this patient progressed further with lung and liver metastases four months after treatment (Fig. 1i–l).

### Apoptosis in zAvatars predicts patient clinical response to treatment

By plotting the zAvatar average induction of apoptosis of each patient (expressed as fold change-FC), we could observe that zAvatars derived from patients with no-progression exhibited a significant higher induction of activated caspase3 upon treatment, compared to zAvatars derived from patients that progressed (Fig. 2a, $p < 0.0001$). In contrast, tumor shrinkage in zAvatars did not predict the absence or presence of disease progression (Supplementary Fig. 4), possibly due to the very fast assay which may not allow sufficient time for effective tumor clearance to occur.

Upon stratifying the data by tumor stages, a consistent pattern emerged: zAvatars from patients with no-progression show a higher apoptosis FC upon treatment when compared to zAvatars from patients who progressed ($p = 0.0363$ and $p = 0.0099$) (Fig. 2b, c). These findings highlight a robust correlation between apoptosis FC in zAvatars and patient's response to treatment, regardless of tumor stage.

To test the ability of the zAvatar-test to segregate clinical sensitivity and resistance, a Receiver Operating Characteristic (ROC) analysis was performed using the average apoptosis fold change (Fig. 2d). The area under curve (AUC) was 0.839 ($p < 0.0001$), and a cut-off value of 1.34 was identified as the optimal threshold, exhibiting the highest levels of specificity and sensitivity (Fig. 2e). In other words, zAvatars with an apoptosis fold induction above 1.34

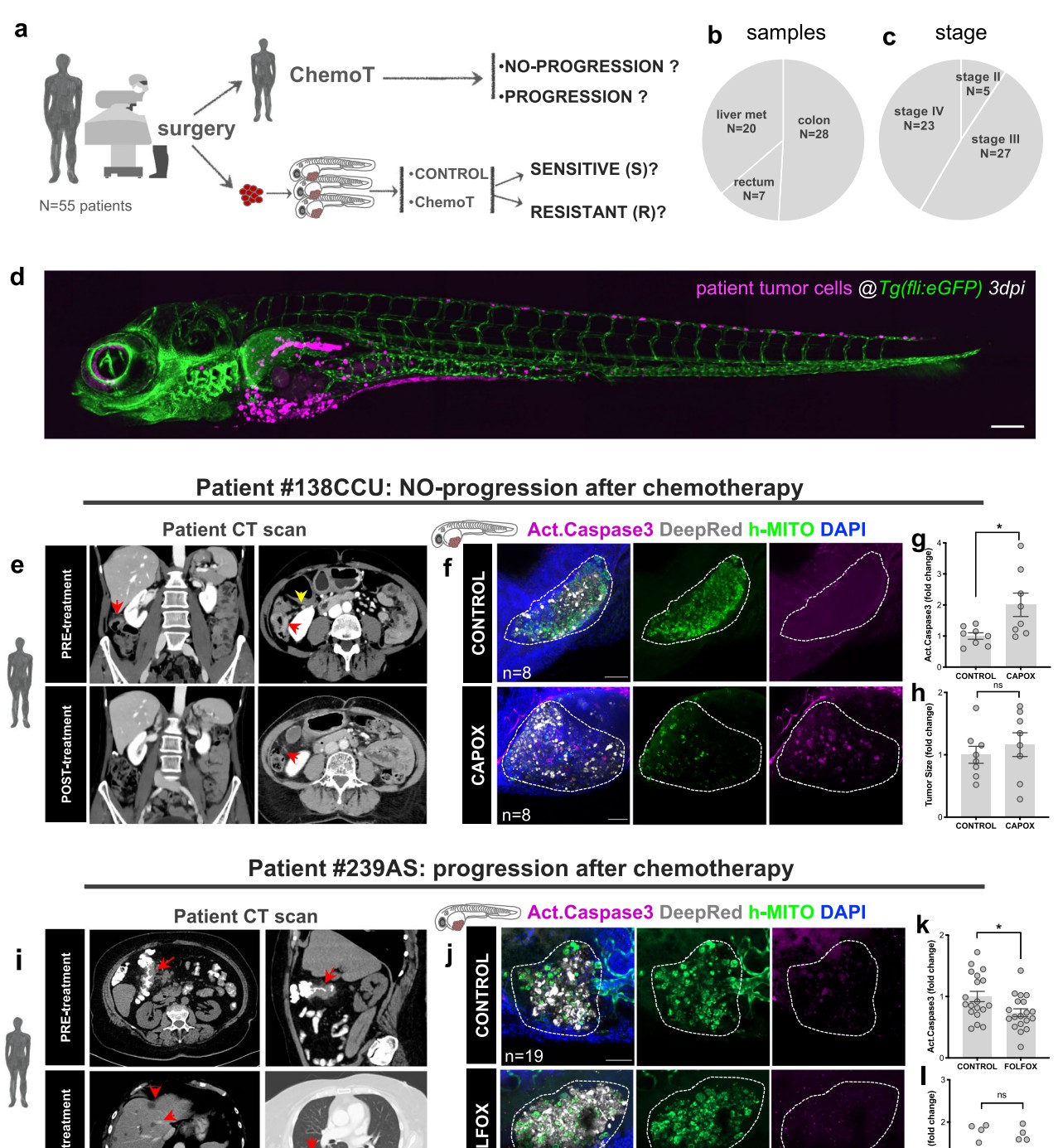

**Patient #138CCU: NO-progression after chemotherapy**

**Patient #239AS: progression after chemotherapy**

were classified as sensitive (S) to therapy, whereas those with a value below or equal 1.34 were categorized as resistant (R) to therapy. Considering this cut-off value, Fig. 2f illustrates a confusion matrix displaying the number of patients with actual and predicted responses in zAvatars, i.e., no-progression/progression patients and sensitive (S) and resistant (R) zAvatars. Out of the 27 sensitive zAvatar-tests, 25 of their matching patients exhibited no-progression disease, yielding a positive predictive value (PPV) of 92,6%. Conversely, out of the 28 resistant zAvatar-tests, 20 of their matching patients experienced disease progression, resulting in a negative predictive value (NPV) of 71,4%. In summary, the zAvatar-test displays a sensitivity of 76% and specificity of 91% (Fig. 2g).

**Metastatic potential in zAvatars correlates with tumor staging and patient clinical progression**

One of the major advantages of using the zAvatar model is the possibility to easily observe and quantify the incidence of micro-metastases in the whole animal, including the tail, eye or gills, as early as 3dpi (Fig. 3a–f). In order to form micrometastases in the zebrafish embryo, tumor cells must have the potential to undergo several processes, including resisting shear stress within circulation, evading surveillance by the host's innate immune system (live imaging Supplementary Movie 1), extravasating and seeding distant sites. By analyzing the incidence of micrometastases in untreated zAvatars at 3dpi, we can access the potential of patient's tumor cells

**Fig. 1 | Clinical study design, patient cohort and examples of zAvatars analysis.**
**a** Design of the study: zAvatars of 55 CRC patients were generated and treated in vivo with the same therapy of the corresponding patient. Response to treatment was compared between patients and their matching zAvatar. **b** Type and (**c**) stage of CRC samples included in the study. **d** Representative image of a zAvatar at 3dpi. Scale bar represents 200 µm. **e** Axial (left image) and coronal (right image) computed tomography (CT) images from patient#138CCU showing the primary tumor (red arrows) in the hepatic flexure of the colon (yellow arrow). Post-operative follow-up imaging revealed no signs of disease recurrence, representing an example of a patient with no-progression. The red arrow indicates right hemicolectomy. **f** zAvatars#138CCU were generated by injecting colon cancer cells from patient #138CCU into the PVS of 2dpf zebrafish embryos. At 1dpi zAvatars underwent the same therapy as the patient (CAPOX) for 2 consecutive days and then compared to untreated controls. Quantification of apoptosis (activated caspase3) in control and treated zAvatars (**g**, $p = 0.0281$). **i** CT scan from patient#239AS displays irregular wall thickening of the hepatic flexure (red arrows) suggestive of

colon cancer. Post-operative follow-up imaging reveals multiple hypodense lesions on liver parenchyma (left image) and a lung lesion (right image) (red arrows), thus representing an example of a patient who progressed. **j** zAvatar#239AS were generated by injecting colon cancer cells from patient #239AS into the PVS of 2dpf zebrafish embryos. At 1dpi zAvatars underwent the same adjuvant chemotherapy as the patient (FOLFOX) for 2 consecutive days and then compared to untreated controls. zAvatars did not show activation of apoptosis after treatment (**k**, $p = 0.0313$). In both examples, tumor cells are labeled in white (cell tracker DeepRed) and DAPI in blue. Maximum Z projections of human-mitochondria marker are shown in green and activated Caspase3 in magenta. Apoptosis (activated Caspase3) (**g**, **k**) and tumor size (**h**, **l**) were quantified at 3dpi. Results are expressed as AVG ± SEM. Each dot represents one zAvatar and the total number (n) of zAvatars analyzed is indicated in the images. A dashed white line delineates the tumor. Scale bars=50µm. Data were analyzed using unpaired two-sided Mann–Whitney test: (ns) > 0.05, *$p ≤ 0.05$. Source data are provided as a Source Data file.

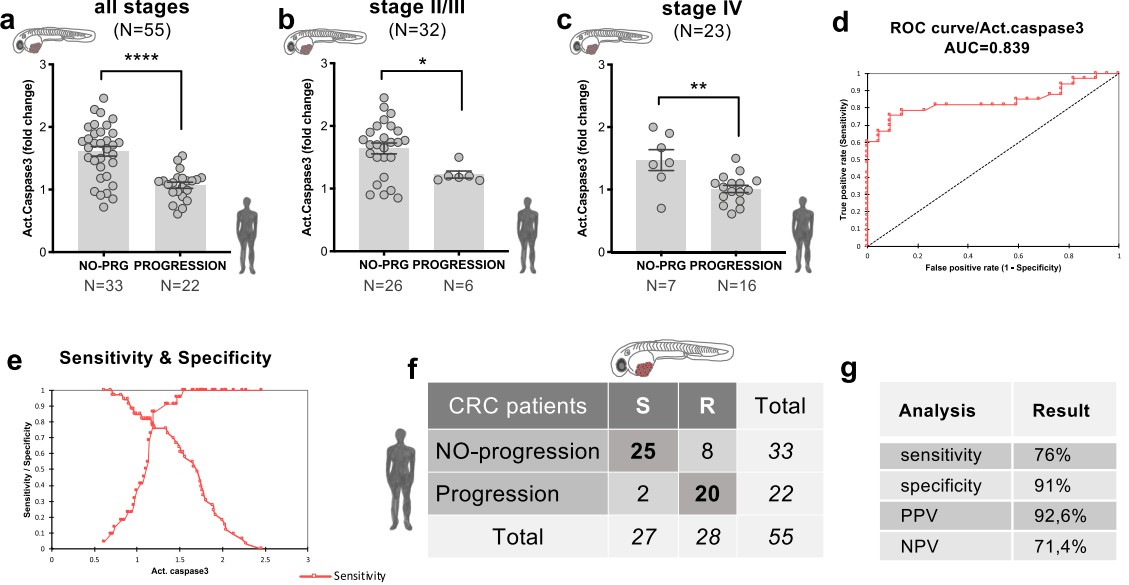

**Fig. 2 | Apoptosis in CRC zAvatars predicts patient clinical response to treatment.** **a** Apoptosis fold change in zAvatars derived from patients with no-progression ($N = 33$ patients, a total of 667 zAvatars analyzed) is significantly higher than zAvatars derived from patients with progression ($N = 22$ patients, a total of 518 zAvatars analyzed); total $N = 55$ patients, $p < 0.0001$. **b** The same trend was observed in stage II/III patients: $N = 26$ patients with no-progression (530 zAvatars analyzed) $vs$ $N = 6$ patients with progression (137 zAvatars analyzed); total $N = 32$ patients, $p = 0.0363$; and (**c**) in stage IV patients: $N = 7$ patients with no-progression (137 zAvatars analyzed) vs $N = 16$ patients with progression (381 zAvatars analyzed); total $N = 23$ patients, $p = 0.0099$. Results are expressed as AVG ± SEM. $N =$ number of patients. Data were analyzed using unpaired two-sided Mann–Whitney test: (ns)

> 0.05, *$p ≤ 0.05$, **$p ≤ 0.01$, ****$p ≤ 0.0001$. **d** ROC analysis of the average fold change of apoptosis for both no-progression ($N = 33$ patients) and progression patients ($N = 22$ patients) of all 55 patients. The area under the curve was 0.839, supporting the ability of the zAvatar-test to discriminate no-progression from progression patients. **e** A cut-off value of 1.34 was identified as the optimal threshold, with 91% specificity and 76% sensitivity ($N = 55$ patients). **f** Confusion matrix displays the number of patients with actual and predicted responses in zAvatars, i.e., sensitive (S) are zAvatars whose fold induction of apoptosis was >1.34, while zAvatars with fold induction of apoptosis ≤1.34 are classified as resistant (R). **g** Values for sensitivity, specificity, positive predictive value (PPV) and negative predictive value (NPV). Source data are provided as a Source Data file.

to go through these metastatic processes[13,26–29], and relate it to tumor stage.

We observed that most zAvatars from stage II/III patients had no micrometastases, whereas in stage IV patients, most of their zAvatars formed micrometastases (Fig. 3g). Nevertheless, we have encountered some exceptions. In stage II/III some patients relapsed, correlating with their high metastatic potential, whereas others responded very well to chemotherapy. This suggests that although the patients' cells had metastatic potential (possible circulating tumor cells and residual disease), adjuvant chemotherapy effectively mitigated that potential. In stage IV, the discrepancy between staging and metastatic potential could be due to the heterogeneity of the sample to which we had access (clones with low metastatic potential). Another hypothesis is that these tumor cells were already in the process of partial MET

(mesenchymal to epithelial transition) rather than EMT (epithelial to mesenchymal transition), and therefore assumed a less invasive, more stable/epithelioid behavior[30,31].

In addition, we have found that zAvatars from patients who experienced progression after treatment present a higher number of micrometastases per zAvatar (Fig. 3h, $p = 0.016$) and higher overall incidence of micrometastases than zAvatars from patients with no progression (Fig. 3i, $p = 0.0055$).

## zAvatars can reveal patient intra-tumoral heterogeneity

To assess if the model has the ability to reveal intra-tumoral heterogeneity, i.e., phenotypic differences between the primary tumor and its metastatic sites within the same patient, we generated zAvatars from patients who underwent synchronous surgeries (Fig. 4). Although

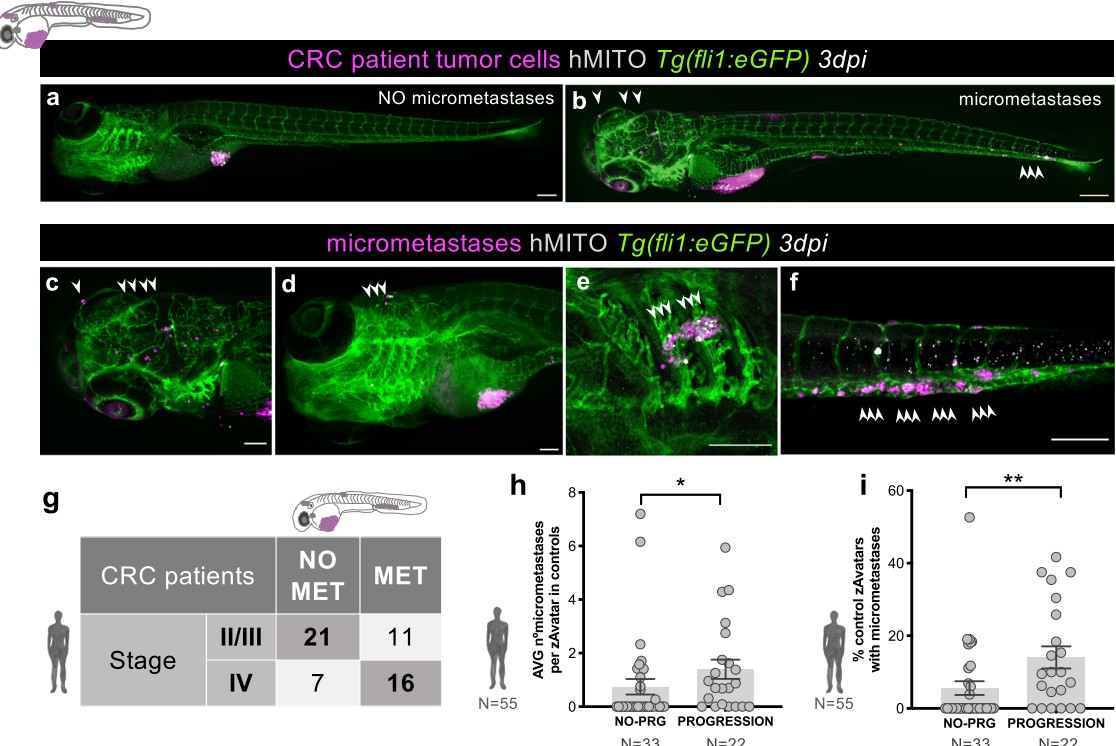

**Fig. 3 | Metastatic potential in zAvatars correlates with tumor staging and patient clinical progression. a** Example of a zAvatar without micrometastases and (**b**) example of a zAvatar with micrometastases, indicated by the white arrows. Scale bars represent 200 µm. Magnification of several examples of micrometastases in the brain (**c**, **d**), gills (**e**) and caudal hematopoietic tissue (CHT) (**f**) at 3dpi. Scale bars represent 100 µm. **g** Confusion matrix displays the number of patients with actual and predicted metastases formation in zAvatars, i.e., 16 patients from stage IV presented also micrometastases in the correspondent zAvatar, while 21 patients from stages II and III did not. **h** The number of micrometastases in each zAvatar (untreated controls) was quantified and then the average per zAvatar for each patient was calculated. zAvatars from patients with disease progression ($N = 22$ patients, a total of 763 zAvatars analyzed) exhibited a higher incidence of micrometastases in comparison with zAvatars from patients with no-progression ($N = 33$ patients, a total of 599 zAvatars analyzed); total $N = 55$ patients, $p = 0.016$. **i** The percentage of zAvatars in control showing micrometastases at 3dpi was quantified and zAvatars from patients with disease progression ($N = 22$ patients, a total of 763 zAvatars analyzed) exhibited a higher incidence of micrometastases in comparison with zAvatars from patients with no-progression ($N = 33$ patients, a total of 599 zAvatars analyzed); total $N = 55$ patients, $p = 0.0055$. Results are expressed as AVG ± SEM. $N$ = number of patients. Data were analyzed using unpaired two-sided Mann–Whitney test: *$p \leq 0.05$, **$p \leq 0.01$. Source data are provided as a Source Data file.

synchronous surgeries are rare, we were able to obtain two noteworthy examples.

In the first case, patient#229CCU presented with a rectal adenocarcinoma with liver metastasis (Fig. 4a). zAvatars from both samples were generated in parallel and apoptosis, tumor size, implantation rates and metastatic potential were analyzed at 3dpi (Fig. 4b–e). Both samples were sensitive to FOLFOX treatment, and the patient also responded to treatment. In the second case, patient#189AS was diagnosed with colon adenocarcinoma with liver metastasis (Fig. 4f). A different behavior between the two samples was observed: while zAvatars derived from the primary tumor were sensitive to FUFOL treatment, those derived from liver metastasis displayed resistance (Fig. 4g–j). This patient presented liver progression three months after completing chemotherapy, matching with the results previously obtained from the zAvatar-test. These results highlight how the response of the metastasis plays a dominant role in the overall patient response, as expected.

In conclusion, the use of different samples from the same patient demonstrates the ability of the zAvatar model to reveal intra-patient tumor heterogeneity, characterized by different in vivo phenotypes that correlate with clinical progression.

**The zAvatar-test can be a valuable platform for testing alternative therapy options**
In this section, we provide examples where we tested not only the chemotherapy regimen administered to the patient but also an

alternative option present in treatment guidelines (Fig. 5). Once more, a variety of scenarios emerged: in some cases, the zAvatar-test was sensitive and concordant with the chosen therapy but resistant to the alternative option (Fig. 5a–c); conversely, in other cases the zAvatar-test was resistant to the given treatment but sensitive to the alternative (Fig. 5d–f). In this small sample of 10 patients, 50% exhibited optimal treatment, 20% received a non-effective option, and 30% displayed resistance to both regimens tested (Fig. 5g). These examples highlight the potential of the zAvatar-test to assist in selecting the most suitable treatment for each individual patient. In other words, instead of achieving an optimal treatment in only 50% of cases, a sensitivity-test could increase these numbers up to 70%. Additionally, it allows to identify upfront multi-resistant tumors to then offer the possibility of exploring off-label options.

**Unbiased decision tree algorithm increases the sensitivity of the zAvatar-test**
Aiming to improve the accuracy of the zAvatar-test, we performed an unbiased multivariate analysis using all available patient and zAvatar variables (Supplementary Data 1). A "Two-Step cluster" analysis identified patient's tumor stage, zAvatar-apoptosis FC, and zAvatar-metastatic potential as the most important factors for predicting patient response (Supplementary Table 3 and Supplementary Fig. 5). Next, a multivariate classification analysis was computed using the "Classification and Regression Trees" (CRT) algorithm (Fig. 6a, b). Once again, the decision tree model identified the same three variables

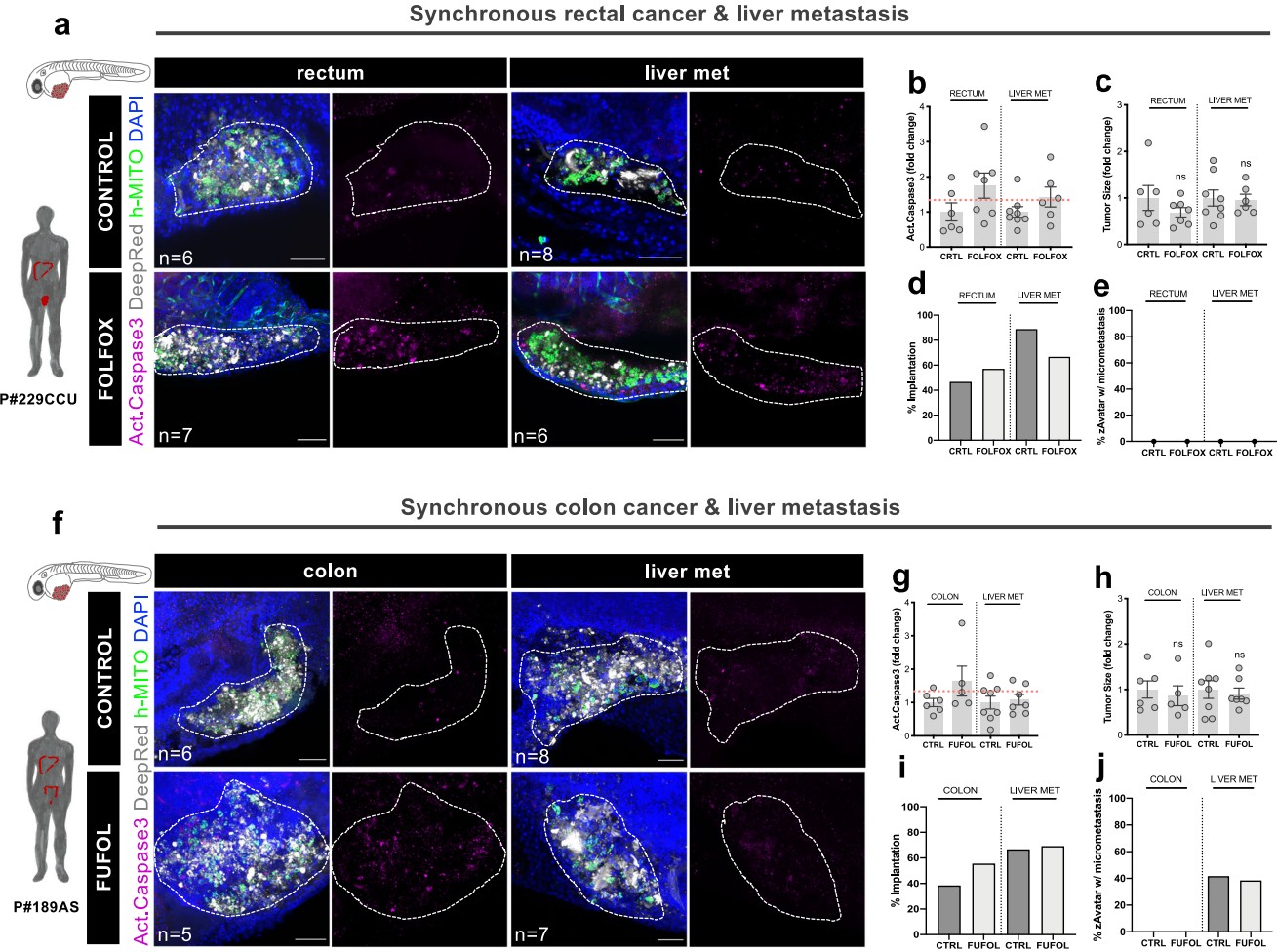

**Fig. 4 | zAvatars can reveal patient intra-tumoral heterogeneity. a** zAvatars were generated from a synchronous surgery of rectal cancer & liver metastasis (patient#229CCU), treated with FOLFOX and compared to untreated controls. **f** zAvatars were generated from a synchronous surgery of colon cancer & liver metastasis (patient#189AS), treated with FUFOL and compared to untreated controls. At 3dpi, apoptosis (**b**, **g**), tumor size (**c**, **h**), implantation rates (**d**, **i**) and metastatic potential (**e**, **j**) were analyzed. Tumor cells are labeled in white (Deep Red), Activated Caspase 3 in magenta, DAPI in blue and human-mitochondria marker in green. Data is expressed as AVG ± SEM. Each dot represents one zAvatar and the total number (*n*) of zAvatars analyzed is indicated in the images. Dashed white line in the images is delimitating the tumor of each zAvatar. Scale bars represent 50 μm. Dashed line in the graphs is marking the 1.34 threshold defined by the ROC curve (**b**, **g**). Data were analyzed using unpaired two-sided Mann–Whitney test: (ns) > 0.05. Source data are provided as a Source Data file.

as robust predictors of patient outcomes and categorized them in a hierarchical order: firstly, it sorted patients by tumor stage (stage II/III vs IV); then, within early stages, by the presence/absence of micro-metastases in untreated control zAvatars (zAvatar-metastatic potential); and finally, by their zAvatar-sensitivity to therapy (zAvatar-apoptosis FC) (Fig. 6a).

The model also revealed distinct apoptosis cut-off values dependent on tumor stage: 1.47 for stage II/III tumors and 1.18 for stage IV tumors (Fig. 6a, b), which we confirmed independently through a ROC curve analysis (Supplementary Fig. 6). Although we do not have an explanation for these two different cut-off values, we may speculate that metastatic tumor cells, known for their increased resistance to treatment, may require only a minor sensitivity to produce a clinically significant effect.

In the context of stage II/III, the metastatic potential variable emerges as a critical factor to improve the zAvatar-test accuracy. Here, patients whose zAvatars had no micrometastases are immediately classified as having no-progression disease. This suggests that, in such cases, sensitivity to therapy is irrelevant for progression outcome, suggesting that these patients may be spared from chemotherapy and its toxic side effects. In contrast, patients whose zAvatars developed micrometastases, are further categorized according to sensitivity to

therapy i.e., the apoptosis FC value (Fig. 6a, b). In contrast, for stage IV patients, the metastatic potential becomes irrelevant, since tumor cells are already metastatic, and now the determining factor is response to therapy.

By applying this refined decision tree to all patients, with just 3 parameters, the PPV changes from 92.6% to 91% but the NPV increases from 71.4% to 90%. In other words, the zAvatar-test successfully anticipated clinical outcome in 50 out of 55 patients, resulting in an overall accuracy of 91% (Fig. 6c, d). Detailed correlations between individual patients and their corresponding zAvatar-test are provided in Supplementary Data 2.

Moreover, this unbiased analysis highlights that in early stages, the metastatic potential of tumor cells is a dominant factor in the prognostic of these patients; in other words, the absence of micro-metastases in the untreated zAvatars may be a favorable prognosis.

In summary, the accuracy of the zAvatar-test improves considerably when considering not only the response to chemotherapy but also the biological characteristics of the tumor cells, such as their potential to form micrometastases and the original tumor staging. This comprehensive approach provides a more robust framework for evaluating patient outcomes and facilitates the identification of individuals who are likely to respond favorably to specific treatments.

## P#41CCU:no-progression after adjuvant treatment

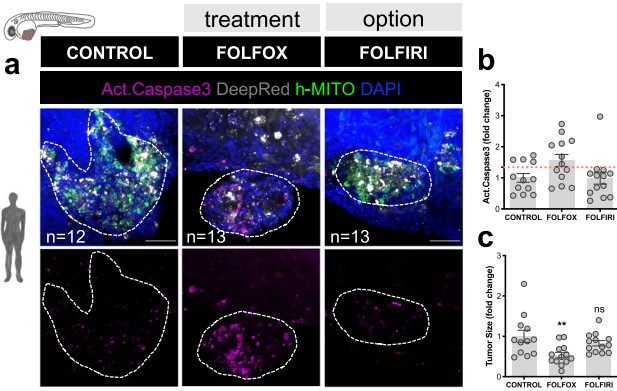

## P#64CCU:progression after adjuvant treatment

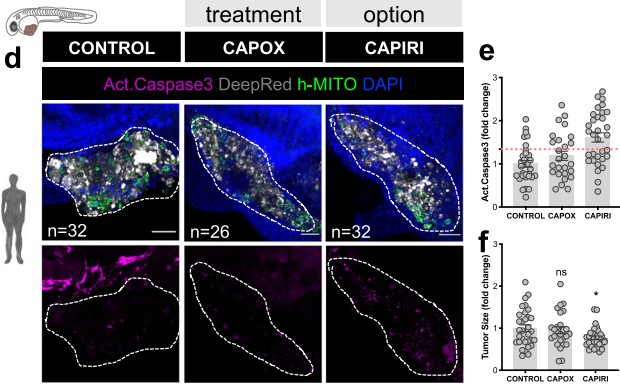

**g**

| zAvatar | patient treatment | outcome | option |
|---|---|---|---|
| 41CCU | FOLFOX | NO-PRG | FOLFIRI |
| 64CCU | CAPOX | PROG | CAPIRI |
| 203CCU | FOLFIRI | PROG | CAPO + Cetuxi |
| 256CCU | CAPOX | NO-PRG | CAPIRI |
| 294CCU | CAPOX | NO-PRG | FOLFIRI |
| 296CCU | CAPIRI | PROG | CAPOX |
| 61AS | FOLFIRI | PROG | FOLFOX |
| 95AS | FOLFOX | NO-PRG | FOLFIRI |
| 110AS | FOLFIRI | PROG | FOLFOX |
| 225AS | FOLFOX | NO-PRG | FOLFIRI |

**Fig. 5 | The zAvatar-test can be used to test alternative therapy options.**
**a**–**c** zAvatar#41CCU was treated in vivo with FOLFOX (chemotherapy regimen) and FOLFIRI (alternative regimen) and compared with untreated controls.
**d**–**f** zAvatar#64CCU was treated in vivo with CAPOX (chemotherapy regimen) and CAPIRI (alternative regimen) and compared with untreated controls. For all examples, apoptosis (**b**, **e**) and tumor size (**c**, **f**) were analyzed at 3dpi. Tumor cells are labeled in white (Deep Red), activated Caspase 3 in magenta, DAPI in blue and human-mitochondria marker in green. Data is expressed as AVG ± SEM. Each dot represents one zAvatar and the total number (*n*) of zAvatars analyzed is indicated in the images. Dashed white line is delimitating the tumor of each zAvatar. Scale bars represent 50 μm. Dashed line in the graphs is marking the 1.34 threshold defined by the ROC curve (**b**, **e**). Data were analyzed using unpaired two-sided Mann–Whitney test: (ns) > 0.05, *$p = 0.0403$, **$p = 0.0082$. **g** Examples of zAvatars tested with the chemotherapy taken by the patient, their clinical outcome and the alternative option. Blue depicts sensitivity and red resistance to treatment. Source data are provided as a Source Data file.

### Patients with a sensitive zAvatar-test have longer PFS
Lastly, to evaluate whether the zAvatar-test translates into a clinical impact on patients' time to recurrence, we plotted the progression-free survival (PFS) curves of patients with a sensitive zAvatar-test *vs*

patient with a resistant zAvatar-test (Fig. 7a). PFS was calculated from the initiation of chemotherapy treatment until the date of last observation or date of disease progression. Remarkably, the mean PFS (mPFS) of patients belonging to the sensitive zAvatar-test group was 3 times longer than that of the zAvatar-test resistant group ($N = 55$, $p < 0.0001$) (Fig. 7b). This trend remains consistent when the analysis is further stratified by stage ($p < 0.0001$ and $p = 0.0063$) (Fig. 7c, d).

In all, our results indicate that CRC patients whose zAvatar-test show sensitivity to treatment experienced a significantly longer PFS compared to those patients who did not respond.

## Discussion
Treatment guidelines in cancer rely on large clinical trials and average response rates, but there is an urgent need for a functional test to assign optimal therapy to individual cancer patients. The zAvatar model is becoming a fast and sensitive in vivo screening platform for personalized medicine, and promising results have been achieved using mainly cancer cell lines or a limited number of patient samples[13,18–25].

Here, we present the results of the largest clinical study with zebrafish Avatars. The zAvatar-test demonstrated a sensitivity of 94% and specificity of 86%, achieving 91% of PPV and 90% of NPV.

The zAvatar model offers the unique advantage of assessing the metastatic potential of patient's tumor cells. Our data showed that not only the incidence of micrometastases correlates with tumor stage, but also that patients who experienced disease progression exhibited a higher incidence of micrometastases in their corresponding zAvatars. These results reveal how the intrinsic biological metastatic potential of tumor cells can serve as a prognostic factor.

This was further confirmed by a multivariate analysis, which in an unbiased manner identified the zAvatar-metastatic potential as a major prognostic factor in early-stage patients. By employing a decision tree model that takes into account tumor stage, zAvatar-apoptosis fold change and zAvatar-metastatic potential, we could improve the overall accuracy of the zAvatar test and successfully predicted clinical outcomes in 50 out of 55 patients.

Importantly, progression-free survival (PFS) was significantly longer in patients whose zAvatar-test was sensitive to treatment, in comparison to patients whose zAvatar-test was resistant.

Fresh sample availability and heterogeneity are probably the main limitations in establishing zAvatars, which are common to all patient-derived models. Mouse and organoid Avatars have shown very similar predictive values[32–38]. However, practical constraints such as time, costs, and the use of Matrigel are associated with these models. Additionally, other emerging models such as 3D spheroids[39,40] and ex-vivo explants[41–43] have also demonstrated very promising results. Nevertheless, these models collectively lack the complexity of an in vivo system necessary for tracking metastatic potential or screening therapies requiring in vivo metabolism, for instance.

Altogether, our results show that the zAvatar-test, with a time-frame of 10 days, has a remarkable predictive value for personalized medicine (Fig. 8). Implementing such an effective test to guide treatment decisions has the potential to transform non-responder patients into responders, even in advanced stages of the disease. Furthermore, it serves as a valuable tool for assessing off-label options, particularly in the case of multi-resistant tumors.

Importantly, this test can be expanded to other types of cancers and therefore may have an impact on the whole oncology field, by optimizing treatment options, improving PFS, preventing unnecessary toxicities and reducing healthcare costs.

In other words, this personalized approach could lead to optimized resource allocation and fewer unnecessary interventions. All this might alleviate financial burden on patients and healthcare systems, contributing to a more sustainable healthcare practice. Lastly, a fast sensitivity test can also help identify eligible patients for clinical

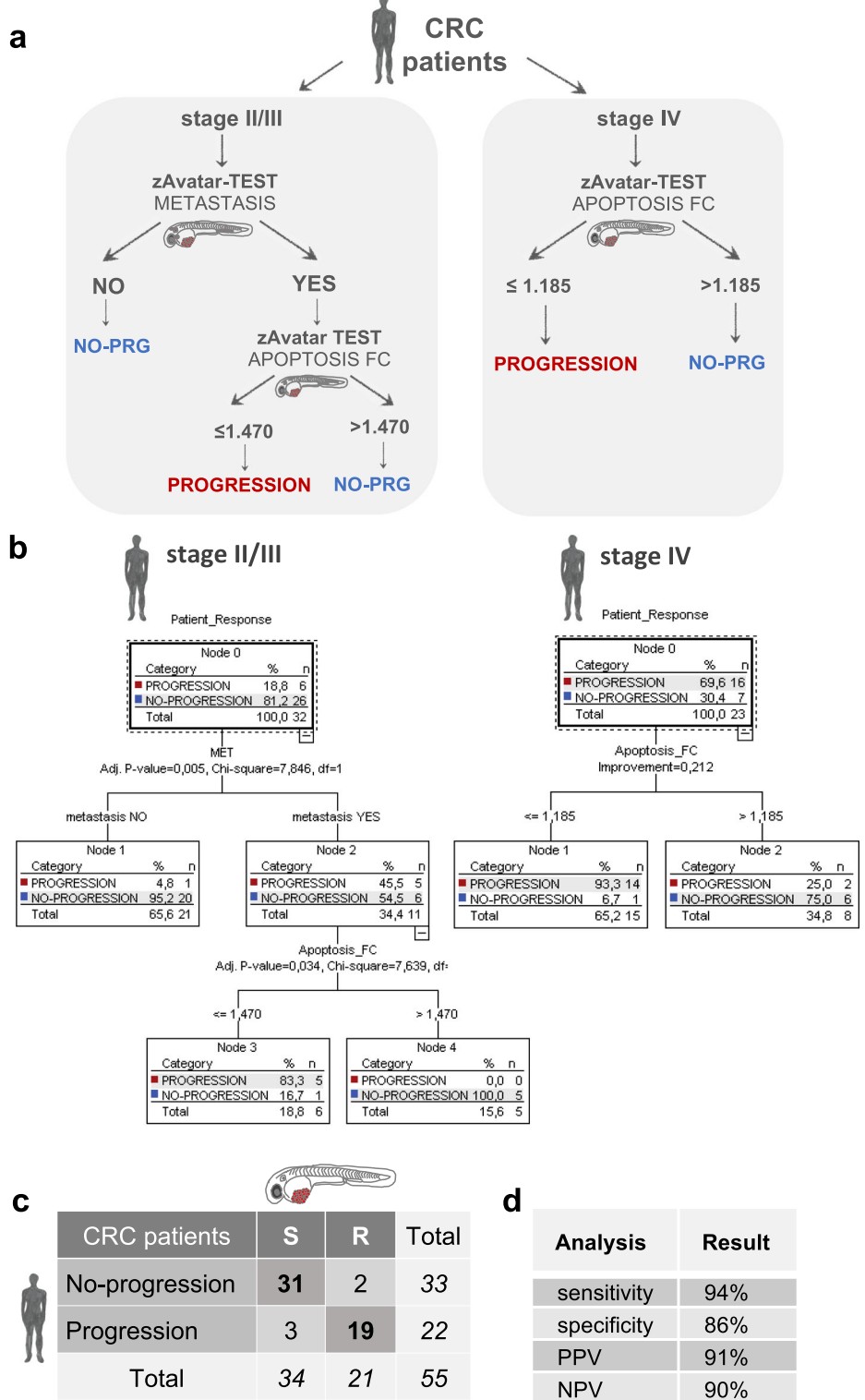

**Fig. 6 | Decision tree model improves accuracy of the zAvatar-test. a** Decision tree model takes into account tumor stage, zAvatar-metastatic potential and zAvatar-apoptosis fold change ("Apoptosis_FC"). **b** The "Patient Response" (Progression/No-Progression) was the dependent variable, while the "Stage Tumor", "Apoptosis_FC" and presence of "Metastasis" were the independent variables. Parent node = 5 cases; child node = 5 cases; overall percentage of correlation = 91%. **c** New confusion matrix according to the decision tree displays the number of patients with actual and predicted responses in zAvatars. Stage II/III sensitive (S) patients refers to patients' whose zAvatars have no metastasis or patients' whose zAvatars present metastasis but the induction of apoptosis is >1.47. Stage IV sensitive (S) patients refers to patients' whose zAvatars fold induction of apoptosis is >1.185. Conversely, stage II/III zAvatars with presence of metastasis and fold induction of apoptosis ≤1.47, along stage IV zAvatars whose fold induction of apoptosis ≤1.185 are classified as resistant (R). **d** New values for sensitivity, specificity, PPV and NPV for the zAvatar-test taking into account the tree decision model. Source data are provided as a Source Data file.

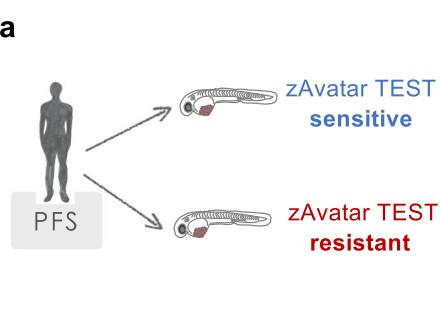

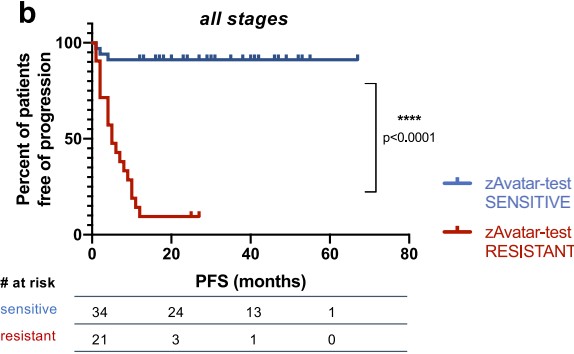

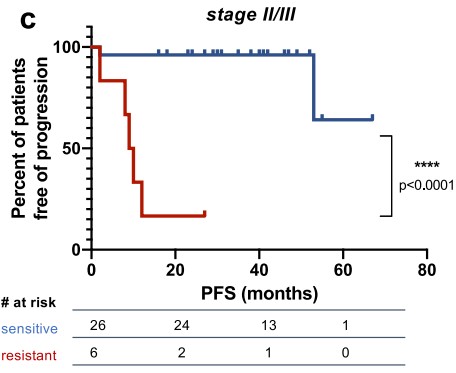

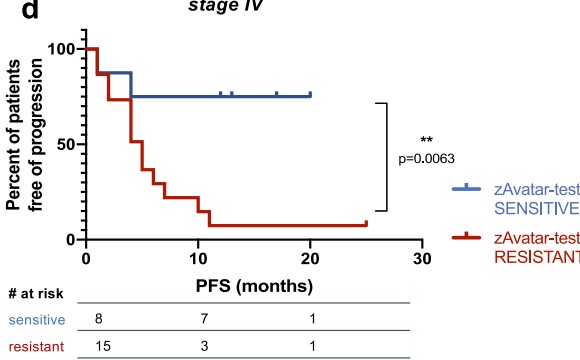

**Fig. 7 | Patients with a sensitive zAvatar-test have longer Progression-Free Survival. a** Kaplan–Meier survival curves were performed comparing the PFS of patients based on sensitivity or resistance of their zAvatar-test (taking into account the tree decision model). The PFS was calculated from the initiation of chemotherapy until either last observation or date of progression. **b** When analyzing patients from all stages, the zAvatar sensitive group had a longer mean PFS of 30.9 months compared to 7.5 months for the resistant group ($N = 55$ patients; $p < 0.0001$). **c, d** Similarly, in stage II/III patients the mean PFS was 37.0 months versus 11.3 months ($N = 32$ patients; $p < 0.0001$), and in stage IV patients the mean PFS was 11.4 months versus 5.9 months ($N = 23$ patients; $p = 0.0063$). Source data are provided as a Source Data file.

trials, to increase their success rates and reduce the high costs that clinical trials entail. To introduce the zAvatar-test into clinical practice, it is crucial to perform a randomized clinical trial comparing zAvatar-based therapeutic decisions with physician's-choice (standard of care), a future step that we are already preparing.

## Methods

### Patient clinical data and follow up
The study was approved by the Ethics Committee of the Champalimaud Foundation and Hospital Prof. Doutor Fernando Fonseca. Surgical resected samples were collected by a dedicated pathologist, after written informed consent. Patients were pseudonymized, and each sample was labeled with a chronological number and the acronym from the hospital of origin. This code does not allow to trace back patient identification. Inclusion criteria: 18 years of age or older patients diagnosed with CRC who underwent systemic chemotherapy after surgery. Patients were treated according to the standard of care based on NCCN/ESMO clinical guidelines decided in a multidisciplinary team (MDT) meeting. Clinical data included standard clinical, surgical and pathological data, KRAS and BRAF status, microsatellite status, and intratumoral infiltrating lymphocytes (Supplementary Table 1). The follow-up data included the type of chemotherapy, number of cycles and duration, and progression-free survival in the 12 months after initiation of chemotherapy. Follow-up of CRC patients was performed according to ESMO and NCCN guidelines, with clinical evaluation every 3 to 6 months[44]. Chemotherapy regimens varied between 5-FU alone, FOLFOX, FOLFIRI or derivatives of these combinations (FUFOL, CAPOX, CAPIRI) and sometimes combined with targeted therapies such as cetuximab or bevacizumab (Supplementary Table 2). Classification of

progression/no-progression disease was based in imagiological findings (CT, MRI, PET), clinical assessment, histological confirmation, all discussed in MDT meetings.

In stage II/III patients, a clinical response to treatment was classified as "no-progression within 12 months" (NO-PRG) if there was no evidence of disease recurrence within 12 months after treatment. On the other hand, progression was defined as recurrence of the disease either at the same site as the primary tumor (local recurrence) or in a distant location (distant recurrence/metastasis), i.e., emergence of new imagiological findings in situ or at distance.

In stage IV patients, a clinical response to treatment was defined as "no-progression" when there was no increase in the remaining disease or evidence of de novo disease (in other words, stable disease). Conversely, progression was defined as an increase of the previous lesions or appearance of new disease during the follow-up period.

Response to zAvatar treatment was blindly compared with the patient's clinical response. Experimental researchers had no previous information about the clinical outcome. After performing the zAvatar-test, results were sent to the physicians for correlation analysis.

### Patient sample processing
Tissue was cryopreserved in 90% (v/v) FBS and 10% (v/v) DMSO until zebrafish microinjection. For microinjection, samples were processed as previously described in[45]. In brief, the tissue is thawed and minced using a scalpel in Mix 1, which is composed of advanced DMEM/F12 (Thermo Fisher Scientific), growth factors, antibiotics, and a ROCK inhibitor (Y-27632, Selleckchem). Subsequently, the tissue fragments undergo mechanical fragmentation pipetting up and down followed by centrifugation (300xg, 5 min, 4 °C). The remaining tissue fragments are enzymatically digested in HBSS

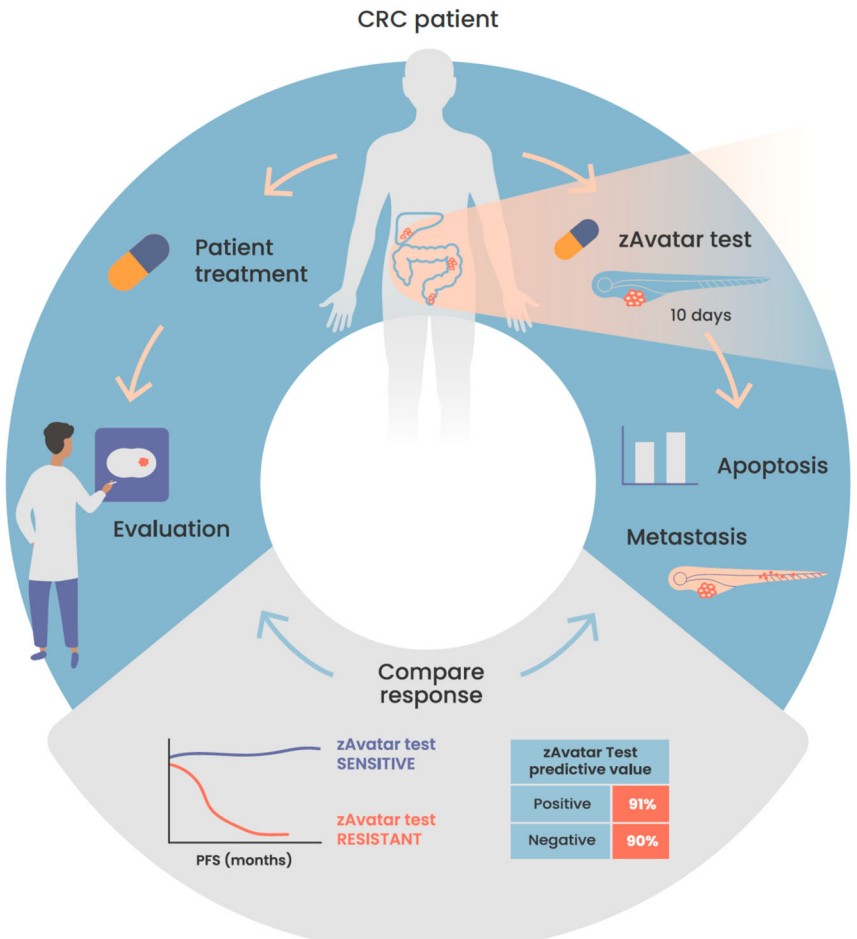

**Fig. 8 | Schematic illustration of the workflow of the zAvatar-test and obtained results.** Our findings demonstrate that the zAvatar-test is an accurate screening-platform for predicting colorectal cancer treatment outcomes. Illustration by Marta Correia.

(Thermo Fisher Scientific) with Liberase (Roche) and DNase I (Thermo Fisher Scientific). Following this step, the fragments are filtered through a 70-μm cell strainer and labeled at 37 °C with a lipophilic dye. Tumor cells are then resuspended in Mix 1 supplemented with 10% (v/v) FBS (Sigma-Aldrich) and checked for viability using Trypan Blue (Sigma-Aldrich) dye exclusion.

### Cell staining
Tumor cells were labeled with Vybrant CM-DiI (Thermo Fisher Scientific) at 4:1000 dilution or with Deep Red (CellTracker, Thermo Fisher Scientific) at 1:1000 dilution (stock 10 μM). Staining was performed according to manufacturer's instructions.

### Zebrafish care and handling
In vivo experiments were performed using zebrafish (Danio rerio), nacre, casper, *Tg(Fli1:eGFP)* and *Tg(mpeg1:mCherry-F)*, which were handled according to European animal welfare Legislation, Directive 2010/63/EU (European Commission, 2016). *Tg(Fli1:eGFP)* allows the visualization of blood and lymphatic vessels, through the expression of eGFP linked to fli1 (endothelial marker) promoter[46]. The Portuguese institutional organizations—ORBEA (Órgão de Bem-Estar e Ética Animal / Animal Welfare and Ethics Body) and DGAV (Direção Geral de Alimentação e Veterinária / Directorate General for Food and Veterinary) have approved this study and its corresponding protocols.

### Zebrafish patient-derived xenograft microinjection
Tumor cells were microinjected into the perivitelline space (PVS) of anesthetized 2 days post fertilization (dpf) zebrafish embryos and maintained at 34 °C until the end of the experiments, as previously described[45,47]. At 1-day post-injection (dpi), zebrafish avatars were screened regarding the presence or absence of a tumoral mass. zAvatars without cells in the PVS, with severe edema or with cell debris were discarded. At 3dpi, zAvatars were sacrificed, fixed with 4% (v/v) formaldehyde (Thermo Scientific) at 4 °C overnight, and preserved at −20 °C in 100% (v/v) methanol (VWR Chemicals).

Percentage of tumor implantation at the end of the assay was calculated as follows:

$$\%\text{implantation} = \frac{\text{n}^{\circ}\text{xenografts at 3dpi with a tumor mass}}{\text{total n}^{\circ}\text{xenografts at 3dpi}} x100 \quad (1)$$

Metastatic potential was quantified based on the percentage of zAvatars that exhibited micrometastases at 3dpi; i.e. zAvatars that presented cells beyond the site of injection (PVS), such as in the gills, tail, or eye. In some cases, these micrometastases could be exclusive to the tail, or they might involve both the tail and gills, for instance.

Metastatic potential was calculated as follows:

$$\text{metastatic potential} = \frac{\text{n}^{\underline{o}}\text{zAvatars with micrometastases in untreated controls @3dpi}}{\text{total n}^{\underline{o}}\text{zAvatars in untreated controls @3dpi}} \times 100$$

(2)

## Drug administration

At 1dpi zAvatars were randomly distributed in the treatment groups: control (E3 medium) and selected chemotherapy, for two consecutive days, and replaced daily. The maximum tolerated concentration (MTC) of anti-cancer drugs was determined using, as a reference, the maximum patient's plasma concentration and testing different doses in non-injected embryos (data not shown). For all drugs, the highest dose without toxic effects was chosen (Supplementary Table 2). Besides the addition to the E3 medium at 1dpi, when patients underwent treatment involving bevacizumab and cetuximab monoclonal antibodies, these agents were added to the cell suspension prior to injection at 100 ng/mL and 20 μg/mL, respectively[13,48].

## Whole-mount immunofluorescence

Primary antibodies: anti-Activated Caspase3 (rabbit, Cell signaling, 1:100, cat#9661), anti-Human mitochondria (mouse, Merck Millipore, 1:100, cat#MAB1273). Secondary antibodies: anti-mouse DyLight 488 (Thermo Fisher Scientific, 1:400, cat#10688674), anti-rabbit DyLight 650 (Thermo Fisher Scientific, 1:400, cat#84546), anti-rabbit DyLight 594 (Thermo Fisher Scientific, 1:400, cat#10108403) were applied simultaneously with DAPI (Sigma-Aldrich, cat#10236276001). zAvatars were mounted with Mowiol aqueous medium.

## Imaging and quantification

Images from tumors in zAvatars were acquired in a Zeiss LSM 710 fluorescence and BC43 Andor confocal microscopes with 5 μm interval z-stacks. Images were analyzed using ImageJ software, using the Cell Counter plugin. The tumor size (number of tumor cells), percentage of activated caspase 3, and percentage of micrometastases were quantified manually by counting all cells in every slice of the tumor (from Zfirst to Zlast)[45,47].

## Statistics and reproducibility

Statistical analysis was performed using GraphPad Prism, IBM SPSS and XLStat software. zAvatars quantification datasets were challenged by normality tests (D'Agostino & Pearson and the Shapiro-Wilk). Data with assumed normal distribution were analyzed by unpaired t-test. Datasets without known distribution were analyzed by the Mann–Whitney test. $\chi^2$ test was performed to test associations with categorical variables. Whenever a value suggestively deviating from the dataset's mean was observed, a comprehensive examination of the dataset was conducted using the "GraphPad Outlier" tool (https://www.graphpad.com/quickcalcs/Grubbs1.cfm).

Receiver operating characteristic (ROC) analysis was performed using XLStat software, considering response to treatment (no-progression disease) as a positive event. Kaplan–Meier curves were performed using GraphPad Prism software and compared with the log-rank test. A multivariate classification analysis was computed using the "Two-Step Clusters" and "Classification and Regression Trees" (CRT) algorithm with IBM SPSS Statistics version 28. The independent variables were selected according to the significant association with "Patient Response", in the bivariate analysis context. Using the CRT growing method with non cross-validation, a five minimum cases in parent and child nodes with tree depth of two were specified. This methodology presents the outputs under a decision tree configuration. Being a dependency technique, and considering the patient's response as an endogenous variable, this method intends to detect the categories of each predictor (tumor stage, apoptosis fold induction, and presence of micrometastases in zAvatars).

For all the statistical analysis, p value is from a two-tailed test with a confidence interval of 95%. Statistical differences were considered significant whenever $p < 0.05$ and statistical output was represented by stars as follows: non-significant (ns) > 0.05, $^*p \leq 0.05$, $^{**}p \leq 0.01$, $^{***}p \leq 0.001$ and $^{****}p \leq 0.0001$. All graphs presented the results as average (AVG) ± standard error of the mean (SEM).

The majority of experiments were performed only once for each patient due to the limited amount of human sample available. Nevertheless, for certain samples, it was possible to cryopreserve more than one vial and indeed we were able to repeat experiments and obtained similar results. This was observed in the cases of zAvatar #61AS, #110AS, #136CCU, and #139CCU.

## Reporting summary

Further information on research design is available in the Nature Portfolio Reporting Summary linked to this article.

## Data availability

Data generated in this study are available within the Article, Supplementary Information or Source Data file. Other data related to this work, including datasets on the fold change of apoptosis, tumor size, and metastatic potential from all zAvatars analyzed across the 55 patients, as well as confocal images ($n = 1245$ zAvatars), are available from the corresponding author upon request. Source data are provided with this paper.

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

## Acknowledgements

This work was funded by the Champalimaud Foundation, FCT-PTDC/MEC-ONC/31627/2017, Congento (LISBOA-01-0145-FEDER-022170, co-financed by FCT/Lisboa2020) and BIAL Award in Clinical Medicine. We thank the Champalimaud Fish Platform and the Surgery Departments and Pathology Services at the Champalimaud Clinical Center and Hospital Prof. Fernando Fonseca. We acknowledge Nica Borgese, Domingos Henrique, Pedro Moura Alves, and Sérgio Simões for the critical reading of the manuscript and Marta Correia for the illustration of Fig. 8. We are also grateful to Miguel Godinho Ferreira and Nuno Figueiredo for the initial support of this project. Finally, we express our sincere appreciation to all the patients who participated in the study.

## Author contributions

R.F. conceptualized and supervised the research. B.C. performed and analyzed all research. M.F.E., V.P., and M.F. performed research. A.Go, L.M.F., J.M.A., I.H., S.B., and C.Carvalho participated in selecting and analyzing patient data. L.M.F., J.M.A., A.A., A.Ga, M.C.M., C.Carneiro, A.P., and V.N. provided patient tumor samples. B.C. and R.F. wrote the paper. A.M. and A.Go supervised all statistical analysis. All authors contributed with the critical reading of the manuscript. All authors have read and agreed to the published version of the manuscript.

## Competing interests

The authors declare no competing interests.
