## [Peer Review File · Nature Communications]

Zebrafish Avatar-test forecasts clinical response to chemotherapy in patients with colorectal cancerREVIEWER COMMENTS

Reviewer #1 (Remarks to the Author): with expertise in zebrafish, cancer

In this manuscript, Costa and colleagues describe the use of the larval zebrafish xenograft platform as a preclinical prediction tool for therapeutic decision-making in colorectal cancer (CRC). Prospective testing of the same treatment regimen in larval PDX avatars (zAvatar) as that used in 55 patients employing apoptosis as a readout (measured by activated caspase 3) demonstrated a positive predictive value of 92.6% and a negative predictive value of 71.4% for response. In addition, the platform was exploited to detect metastatic tumor behavior, heterogeneous primary and metastatic site differential responses to the same therapy and predict progression free survival at all stages of CRC based on the cognate zebrafish larval PDX sensitivity to chemotherapy.

While this paper represents the largest sample size to date for applying the larval zebrafish xenograft model as a personalized cancer therapy prediction algorithm and the responses observed overall are highly correlative and encouraging, there remain a number of concerns to be addressed.

Major:

Using apoptosis as a readout, only 45 of 55 samples showed similar response in the zebrafish and the patient. How do the authors account for the inconsistency in these 10 samples? Should a secondary cell-based readout be incorporated that potentially could further increase accuracy and ultimately sensitivity?

For all the zebrafish images presented, most human cancer cells (DeepRed labeled) do not appear to be h-MITO positive. DAPI positivity is also difficult to visualize.

Figure S2 – was the impact on angiogenesis evaluated as part of the therapeutic responses for any of the patients? If not, this figure is really non-contributory to the story and mention of evaluation of angiogenesis should be removed from the text. If the authors want to include this figure, they need to show an uninjected larva as a control to demonstrate

baseline vasculature in the absence of tumor cell transplantation and evidence of a treatment impact on angiogenesis.

Line 272-275: The authors state that tumor size/shrinkage did not predict response, which is attributed to the speed of the assay. This needs to be clarified. One would expect even if the tumor cells were not all cleared there would be a reduction in DeepRed cells or h-MITO positive cells, which could be quantified.

Additional clarification is needed around the micrometastasis analysis. How was micrometastasis quantified/defined? Is it based on the number of cells that have migrated out of the PVS or if cells migrated to multiple sites at one time? It would be helpful to see images of the same fish at 1 dpi and 3dpi to confirm there were no cells present in the brain, gills, or tail at this earlier time point.

By definition, patients with Stage 2/3 disease do not have metastases, the presence of which would classify them as Stage 4. Thus, while the high degree of correlation of micrometastases in zAvatar transplanted with Stage 4 tumors is encouraging, there were still 11 early-stage patients who demonstrated micrometastases in zebrafish and 7 Stage 4 patients without metastases. While the authors later incorporate this as part of their decision tree analysis, they should account here for the lack of clinical correlation. Is the assumption that those lower stage patients that show micrometastases in their zAvatar have clinically undetected metastases that account for their poor response? What about those Stage 4 patients that don't have corresponding metastases in their zAvatar – do they have clinical metastases that don't show tumor heterogeneity and are responsive to chemotherapy? This is not explicitly explained.

The authors have included the sensitivity threshold of 1.34 in Figure 5 in the assessment of alternative therapies. While this thresholding seems reasonable in the initial studies to define response, justification for continuing to use this bar as a measure for additional or different therapies needs to be provided. This threshold may limit the ability to detect more incremental responses, which may still be of significant clinical value particularly in the context of personalized treatment for high risk patients.

Fig 5c/legend: The table only lists zebrafish response (sensitive/resistant) to each treatment. It would be beneficial to include an additional column with the patient response matched to each zAvatar.

Line 403-417: "In the context of stage II/III, the metastatic potential variable emerges as a critical factor to improve the zAvatar-test accuracy. Here, patients whose zAvatars had no micrometastases are immediately classified as having no-progression disease. This means that their sensitivity to therapy is irrelevant for progression outcome, suggesting that these patients may be spared from chemotherapy, and its toxic side effects." While personalization of cancer therapy is desired, as is a reduction in side effects, particularly if the drug is ineffective, the conclusion here is confusing. Those patients with Stage 2/3 disease responded to therapy as predicted by their zAvatar, so why eliminate chemotherapy for this population? This does not appear a sound recommendation based on the data presented. It would be more helpful to predict for which patients chemotherapy is ineffective, as this population would be subjected to risk of toxicities without clinical benefit. As such, they should not be given chemotherapy, but rather an alternative treatment be considered.

Methods: Graphs display the average of fold changes in Caspase 3 staining – this may not be the optimal way to analyze this data. Moreover, cells are counted manually, which can lead to inaccuracies, particularly when trying to evaluate in 2-D something that exists in 3-D. Authors should comment on these approaches and consider complementing with additional analyses to confirm consistency in their findings.

Minor:

"With exception of some success cases, current cancer molecular and genetic biomarkers have proven insufficient when it comes to reliably predicting treatment outcomes. Most cancer patients do not benefit from genomic precision medicine due to a combination of factors, including the absence of targetable mutations, the lack of effective drugs for specific promising targets and also the possible genetic interactions that may occur between different tumor subclones or with the tumor microenvironment (9, 10)." This comment

undersells clinical advances that have been achieved in some cancer types through molecular profiling and moreover the zAvatar system provides an in vivo platform for functional validation of molecularly targeted therapeutics. The language should be changed to better reflect these realities.

Results, lines 231-232, Please clarify what is meant by the word "blindly" here.

Figure 1D – the cross-sectional CT images of patient#138 are not at the same anatomic level pre- and post-treatment, making correlation regarding response difficult to assess.

Figure 1G' -there is significantly more apoptosis in the control group compared to the treated group? Can the authors explain? Are cells naturally dying?

Figure 4, d and i, need to define "implantation rates".

Results, lines 345-346- "This patient presented liver progression three months after completing chemotherapy, matching with the results from the liver metastasis zAvatar-test." language needs to be changed here as it sounds as if there was a 3-month ZF study conducted in parallel.

Consider moving Results Section 5 "alternative therapies" to the end of the Results section following the Decision Tree analysis to improve the flow of the paper.

Figure 6 legend (line 426) should be (c) instead of (a).

Methods: Whole mount immunofluorescence: Are both secondary antibodies anti-rabbit (for Caspase3) and anti-mouse (for h-MITO) Alexa 488 (green)? Was the anti-mouse 647 used for h-MITO when injecting cells into the fli1a:eGFP?

Reviewer #2 (Remarks to the Author): with expertise in zebrafish, cancer

The manuscript from Costa et al., "The zAvatar-test forecasts patient's treatment outcome

in colorectal cancer: a clinical study towards personalized medicine," demonstrates the utility of zebrafish patient-derived xenograft models (zAvatar) as a fast predictive platform for personalized treatment in colorectal cancer. The major strength of this research is that it, to date, has performed the most extensive co-clinical study of the link between various data points from patient-derived zebrafish xenograft and overall patient outcomes and showed several properties of the PDZX are statistically linked with patient outcome. The methodology of the study is sound, and the conclusions are well supported by the data. This manuscript is significant in the oncology field, as the zAvatar model may hold the potential to enhance personalized medicine by providing clinicians with additional data sets that can be used to optimize treatment options for each patient.

Overall, the manuscript is well-written and straightforward. Some minor issues should be clarified during the revision process:

- It will be helpful to state in the methods or results the degree of blinding that is carried out in this study. For example, were researchers who carried out the zAvatar test blinded to the type of patient sample? What about the team that classified progression/no progression disease?
- Were the patient cells always injected at the same concentration of cells? If yes, please state how many cells were injected in the methods, and if not, please explain how cell numbers were chosen. The number of injected cells seems inconsistent in the figures. For example, in Figure S2, one larvae has many cells, and another has fewer. Please explain. Would the number of injected cells affect the conclusions drawn about angiogenesis, tumor size, and number of apoptotic cells?
- In the methodology (line 599), the authors say that the tumor size and apoptosis were measured by the number of cells. However, the graphs show tumor size and caspase-3 as fold change. Please clarify how the analysis is done.
- The authors should clarify why they labeled both tumor cells (DeepRed) and human mitochondria (h-MITO). I would expect that all injected cells would be double-labeled.

However, in Figures 2, 4, and 5, there are cells labeled with one stain but not the other. Please explain this.

- From the supplemental table, it seems that only 2 patients received bevacizumab or cetuximab. Yet, the methods state that these antibodies were included in the injection mix for all samples. Please clarify why this was done.
- Authors justify that tumor shrinkage in zAvatars did not predict the absence or presence of disease progression, possibly due to the very fast assay that may not allow sufficient time for effective tumor clearance. Why did the authors choose to treat the zAvatars for just two days? Other papers from the same authors typically use a three-day treatment for the zebrafish xenografts. Please explain this inconsistency.
- Do the authors have any explanation why, in some patients, the increase of activated caspase is correlated with a decrease in tumor size (Figure 5) and in other situations, it is not (Figures 1 and 4)?
- In Figure 4, the authors also concluded that both samples of P#229CCU were sensitive to FOLFOX treatment. However, there is no statistical analysis in graph B. In the second case (P#189AS), authors concluded that zAvatars derived from the primary tumor were sensitive to FUFOL treatment, and those derived from liver metastasis displayed resistance. Again, there is no statistical analysis in graph G. Please add the statistics used to draw these conclusions.
- In Figure 7 and the text, it is unclear if the authors used the multivariate analysis threshold to classify sensitive and resistant avatars.
- Overall, some of the text in figures is very small and difficult to see even on zoom, particularly in Fig 2D-E. Other figures are pixelated, such as the graphical abstract and graphs next to images. These should be fixed if possible.

Reviewer #3 (Remarks to the Author): with expertise in colorectal cancer, therapy

1. Enrolled patients included Stage II/III and Stage IV patients, there are many issues arising from this decision

a. Goals of therapy are different for Stage II/III versus Stage IV patients. The endpoint in adjuvant therapy is usually disease recurrence and is usually reported as 3 year or 5 year recurrence free survival. It is not clear whether “no evidence of disease recurrence within 12 months after treatment initiation” can be used as evidence of “no progression”.

b. 6 of 32 Stage II/III patients had recurrence within 12 months. This number seems to be high, compared to those reported in the literature.

2. Pre-treatment and post-treatment CT images were from different areas. For Patient 138CCU, the patient underwent surgical resection for colon cancer. Pre-treatment and post-treatment images were not helpful.

3. Figure 1 E shows n =8, while Figure 1 G shows n = 19, it is not clear why the number of repeats is different. This seems to be the pattern throughout the manuscript, experiments in different patients/different chemo were performed different times.

4. Based on Figure 1, there appears to be significant intra-subject (zebrafish) variation in apoptosis FC. Figure 2D and 2E seemed to be generated from average values.

5. Similarly, Figure 4 B/C/G/H show large intra-subject variability. For example, there appears to be an outlier in Figure 4B, rectum/FOLFOX group. Differences between control and FOLFOX were likely driven by this outlier.

6. 79 patients were enrolled in the study, zAvatar-tests were successful in only 55 patients. Results presented in Figure 2F and 2G only included 55 patients.

7. Since Stage II/III patients have different prognosis compared to Stage IV patients, it is not surprising that tumor stage is identified as a factor in multivariate analysis.

The zAvatar-test forecasts patient's treatment outcome in colorectal cancer: a clinical study towards personalized medicine

Reviewer #1 (Remarks to the Author): with expertise in zebrafish, cancer

In this manuscript, Costa and colleagues describe the use of the larval zebrafish xenograft platform as a preclinical prediction tool for therapeutic decision-making in colorectal cancer (CRC). Prospective testing of the same treatment regimen in larval PDX avatars (zAvatar) as that used in 55 patients employing apoptosis as a readout (measured by activated caspase 3) demonstrated a positive predictive value of 92.6% and a negative predictive value of 71.4% for response. In addition, the platform was exploited to detect metastatic tumor behavior, heterogeneous primary and metastatic site differential responses to the same therapy and predict progression free survival at all stages of CRC based on the cognate zebrafish larval PDX sensitivity to chemotherapy.

While this paper represents the largest sample size to date for applying the larval zebrafish xenograft model as a personalized cancer therapy prediction algorithm and the responses observed overall are highly correlative and encouraging, there remain a number of concerns to be addressed.

We would like to thank reviewer#1 for the critical and careful reading of our manuscript and the opportunity to address all concerns raised, improving our manuscript.

Major:

1. Using apoptosis as a readout, only 45 of 55 samples showed similar response in the zebrafish and the patient.

1a. How do the authors account for the inconsistency in these 10 samples?

Thank you for your comment we will try to clarify.

To address this, we conducted a comprehensive multivariate analysis. We found that by taking into account tumor stage and metastatic potential and by employing a tree decision model, accuracy increased from 82% (45/55) to 91% (50/55) and sensitivity from 76% to 94% (as shown in Fig 2G and Fig 6D).

Furthermore, we explored whether other patient characteristics might contribute to this inconsistency, such as tumor differentiation, microsatellite status, KRAS status, or perineural invasion, among others (Table S2). None of these factors proved to be statistically significant within our sample size.

In other words, among the 10 patients that we could not allocate sensitivity/resistance just based on apoptosis fold change – 5 are now correctly allocated.

What was striking from the model, was that in early-stage patients the metastatic potential becomes dominant; i.e. if the zAvatars do not show metastatic potential, these patients will be assigned to NO-progression, regardless of their apoptosis response. This revelation significantly impacts predictions, meaning that a zAvatar which previously anticipated progression (with an apoptosis induction <1.34), is now reclassified as NO-progression according to the decision tree model because tumor cells did not have metastatic potential.

Moreover, the ROC curve analysis defined different thresholds according to stage – which also changed the proportions. We do not understand why the threshold is so different from early to late stage.

In summary, the partial inconsistency observed with apoptosis as the sole readout can be attributed to the oversight of not considering the biological metastatic potential of the tumor cells and that we were not considering different apoptosis FC thresholds for the different stages.

In the remaining 5 cases that were unpredictable, we can speculate that the inconsistency can be attributed to variations in sample quality and the degree of heterogeneity, both of which can impact on the assay.

1b. Should a secondary cell-based readout be incorporated that potentially could further increase accuracy and ultimately sensitivity?

Thank you for your comment. In fact, we extensively explored various readouts for cell death, such as necrosis, necroapoptosis, autophagy, DNA damage, acridine orange etc. However, in our experience, we could not find any consistently reliable readouts that would correlate to therapy sensitivity as effectively as Activated Caspase 3.

We tried tumor size but also it did not correlate, i.e. we could not detect statistical significant differences (FigS3). Nevertheless, we still integrated tumor size into the multivariate analysis, but it did not show a robust predictive value.

2. For all the zebrafish images presented, most human cancer cells (DeepRed labeled) do not appear to be h-MITO positive. DAPI positivity is also difficult to visualize.

Thank you for your comment. We will try to clarify to the best of our knowledge and experience.

In our experience, labelling human cancer cells with lipophilic dyes (DeepRed or DiI) can be very heterogeneous, even in cell lines, so in patient samples can be even more heterogeneous. Some cells take up better some dyes than others. This heterogeneity is further pronounced in patient samples, where variations in dye uptake among different cells are more evident. Also, we have noted that the brightest cells (and that are better visualized in our images) often are dying cells. To visualize the low intensity staining we would have to overexpose the images.

We use hMITO antibody (MAB1273) as a quality control to confirm the presence of human cells. This is crucial since phagocytes may phagocytose human cells and uptake the dye and therefore emit a signal that is no longer from human tumor cells but actually from zebrafish phagocytes. Therefore, we believe that because of these 2 reasons - heterogeneity of the DeepRed signal and higher intensity in dead cells, coupled with the potential interference from phagocytosis—the 2 signals might not always co-localize as expected.

In addition, the h-Mito antibody (MAB1273) is specifically designed to target the surface of intact mitochondria, meaning that its specificity is directed towards antigens present on the outer surface of mitochondria. The expression levels of mitochondrial antigens can vary not only among different types of cancer cells and also in healthy cells (*Criscuolo et al, 2021, doi: 10.3389/fonc.2021.797265, Chen et al, 2021, https://doi.org/10.1038/s41392-023-01546-w*). Interestingly, when we started working with this hMITO antibody, we observed an unexpected heterogeneity of the hMITO signal in the zAvatars and therefore we performed an immunofluorescence of a patient sample in a paraffin section (**FigR1**). Surprisingly, we also found a huge heterogeneity of hMito in paraffin tumor samples.

Figure R1 – Heterogeneity of hMito staining in paraffin sections of a CRC patient sample

DAPI

Thank you for your comment; again we have been struggling with this as well. We observe a huge heterogeneity in DAPI staining, and when comparing with zebrafish cells, we consistently observe a much less intense signal in human tumor cells (cell lines or primary tumors).

We speculate that human cancer cells, are larger than zebrafish cells and have a less condensed chromatin, affecting DAPI staining intensity. Additionally, human cancer cells have a variety of nuclear morphologies, a characteristic that has been described in the process of transformation.

Furthermore, these cells typically have low mitotic activity and may undergo processes such as senescence or apoptosis, which can contribute to a reduced intensity in DAPI staining.

3. Figure S2 – was the impact on angiogenesis evaluated as part of the therapeutic responses for any of the patients? If not, this figure is really non-contributory to the story

and mention of evaluation of angiogenesis should be removed from the text. If the authors want to include this figure, they need to show an uninjected larva as a control to demonstrate baseline vasculature in the absence of tumor cell transplantation and evidence of a treatment impact on angiogenesis.

Thank you for pointing this out.

We did not compare the presence of angiogenesis in the zAvatar with the patient responses because not all zAvatars were generated in the *Tg(fli1:eGFP)*. Out of the 55 patients only 34 zAvatars were generated in *Tg(fli1:eGFP)*, only 8 zAvatars exhibit angiogenic potential.

Our goal was to show the potential use of this tool for future investigations, particularly in cases where bevacizumab might be a selected treatment. Nevertheless, we agree that without a correlation this information does not contribute to the narrative. Therefore, we have decided to remove this figure.

4. Line 272-275: The authors state that tumor size/shrinkage did not predict response, which is attributed to the speed of the assay. This needs to be clarified. One would expect even if the tumor cells were not all cleared there would be a reduction in DeepRed cells or h-MITO positive cells, which could be quantified.

Thank you for the raised question; we were also surprised by the finding that tumor size did not statistically correlate with clinical outcome.

Again, in our experience, the brightest cells (and that are better visualized in our images) are cells that are in the process of dying and eventually undergo phagocytosis and clearance. It's important to note that, although the Deep Red datasheet indicates that it is retained in live cells, this property is only true at the time of staining. Once a cell is dying, the staining will persist within the cell, as it forms a non-permeant product.

Unfortunately, our quantifications of tumor size based on DeepRed cells/ DAPI double staining did not show statistical significance (FigS3). Despite this, we still incorporated tumor size in the multivariate analysis but once again it did not emerge as variable with predictive value. Nevertheless, we have re-quantified all images using now tumor area based on DeepRed staining and area of hMITO (all slices), again this was not statistically significant (**FigR2**).

Figure R2 – Quantification of tumor area by quantifying area of DeepRed staining (A) and area of hMITO (all slices).

5. Additional clarification is needed around the micrometastasis analysis. How was micrometastasis quantified/defined? Is it based on the number of cells that have migrated out of the PVS or if cells migrated to multiple sites at one time? It would be helpful to see images of the same fish at 1 dpi and 3dpi to confirm there were no cells present in the brain, gills, or tail at this earlier time point.

Thank you for your comment.

Micrometastasis potential was quantified based on the percentage of zAvatars that exhibited micrometastases at 3dpi; i.e. zAvatars that presented cells beyond the PVS, such as in the gills, tail, or eye. In some cases, these micrometastases could be exclusive to the tail, or they might involve both the tail and gills, for instance.

$$\text{metastatic potential} = \frac{n^{\circ}\text{zAvatars with micrometastases in untreated controls @3dpi}}{\text{total } n^{\circ}\text{zAvatars in untreated controls @3dpi}} \times 100$$

We included this explanation in the methods section and changed Fig.3 to be more clear.

We also quantified the number of micrometastases per zAvatar (**FigR3**) but we only used the metastatic potential for our multivariate analysis, since it showed a higher statistical power.

Figure R3 – Quantification of the AVG number of micrometastasis in zAvatars according to patient clinical outcome.

In relation to earlier time points, based on our experience, many tumor cells enter circulation (either actively or experimentally) and at 1dpi is possible to observe many zAvatars with tumor cells in distant sites.

However, as time goes by, these cells tend to disappear, i.e., cells have to survive shear stress of circulation and evade the innate immune system, thus many cells tend to be cleared, and only a small fraction of zAvatars have micrometastases at 3dpi. Therefore, we specifically quantify micrometastases at 3dpi, after the initial clearance. In other words, we only quantify micrometastases that were able to resist shear stress, evade the innate immune system of the host, completed extravasation, and colonized distant sites (i.e performed the last steps of the metastatic cascade).

To access this point, we injected new liver samples that were not included in this study, to show that it is possible to have micrometastases at 1dpi but then at 3dpi they may be cleared, or they can remain (**Fig R4 A, B**). Also, some zAvatars might not have micrometastases at 1dpi but then have at 3dpi (presumably cells took more time to get out of the injection site) (**Fig R4 C**).

Figure R4. New liver samples were injected and zAvatars were imaged at 1dpi and 3dpi. **A.** Example of a zAvatar that shows micrometastasis in the CHT at 1dpi and 3dpi. **B.** Example of a zAvatar that shows micrometastasis in the CHT at 1dpi but not at 3dpi. **C.** Example of a zAvatar that do not show micrometastasis in the CHT at 1dpi but have at 3dpi. White arrows are pointing to micrometastasis.

6. By definition, patients with Stage 2/3 disease do not have metastases, the presence of which would classify them as Stage 4.

Thus, while the high degree of correlation of micrometastases in zAvatar transplanted with Stage 4 tumors is encouraging, there were still 11 early-stage patients who demonstrated micrometastases in zebrafish and 7 Stage 4 patients without metastases. While the authors later incorporate this as part of their decision tree analysis, they should account here for the lack of clinical correlation. Is the assumption that those lower stage patients that show micrometastases in their zAvatar have clinically undetected metastases that account for their poor response? What about those Stage 4 patients that don't have corresponding metastases in their zAvatar – do they have clinical metastases that don't show tumor heterogeneity and are responsive to chemotherapy? This is not explicitly explained.

Thank you for your comment-we apologize for not being clearer.

“By definition, patients with Stage 2/3 disease do not have metastases, the presence of which would classify them as Stage 4.”

Yes correct. Our Stage 2/3 patients were patients classified as stage 2/3 due to lack of detectable distant disease at the time of diagnosis and surgery. These patients were recommended (according to international guidelines) to be treated with adjuvant chemotherapy to reduce the chances of relapse/progression; in other words, these patients have the risk to have undetectable residual disease or circulating tumor cells and therefore need to be treated after surgery to reduce their chances of having a relapse.

Thus, while the high degree of correlation of micrometastases in zAvatar transplanted with Stage 4 tumors is encouraging, there were still 11 early-stage patients who demonstrated micrometastases in zebrafish and 7 Stage 4 patients without metastases. While the authors later incorporate this as part of their decision tree analysis, they should account here for the lack of clinical correlation.

Thank you for your comment but we disagree. We show that the majority of zAvatars from early-stage patients do not form metastasis (21/32) and the majority zAvatars derived from late-stage patients do show micrometastasis (16/23).

However, there are exceptions, as you point out, which we were also puzzled and thought that the metastatic potential was not giving much information, until we performed the unbiased multivariate analysis and the tree decision model.

If we analyze carefully these 11 early-stage patients: 5 patients progressed, aligning with our decision tree results indicating a correlation between the presence of micrometastases and patient progression (64CCU, 100CCU, 171AS, 176AS, 197AS). Another 5 patients responded very well to chemotherapy, suggesting that although the patient's cells had metastatic potential (and possible circulating tumor cells and residual disease), adjuvant chemotherapy effectively mitigated that potential (58CCU, 67CCU, 79CCU, 110CCU, 135CCU).

One patient remains unexplained in this context and was indeed categorized as a non-match (134CCU).

In summary, as the tree decision model highlights, we cannot analyze each variable alone—clinical outcome depends not only on the metastatic potential but also on whether the patient was treated with the right treatment.

11 early-stage patients with micrometastases		
58CCU	stable	caspase 1.74
67CCU	stable	caspase 2.10
79CCU	stable	caspase 1.78
110CCU	stable	caspase 1.70
135CCU	stable	caspase 1.50
134CCU	stable	caspase 0.90
64CCU	progression	caspase 1.20
100CCU	progression	caspase 1.14
171AS	progression	caspase 1.18
176AS	progression	caspase 1.13
197AS	progression	caspase 1.50

Is the assumption that those lower stage patients that show micrometastases in their zAvatar have clinically undetected metastases that account for their poor response?

Yes, clinically undetected metastases, residual disease or circulating tumor cells. This is what our data suggests and also aligns with the fact that these patients were considered for adjuvant chemotherapy. If there were no risk for progression, these patients would have not been treated.

What about those Stage 4 patients that don't have corresponding metastases in their zAvatar – do they have clinical metastases that don't show tumor heterogeneity and are responsive to chemotherapy? This is not explicitly explained.

In Stage 4 patients, the majority of zAvatars exhibit micrometastases (16/23), however we have encountered exceptions. We can speculate that this discrepancy could be due to the heterogeneity of the sample that we had access (only got clones with low metastatic potential). Another hypothesis is that these tumor cells were already in the process of MET (mesenchymal to epithelial transition) rather than EMT (epithelial to mesenchymal transition), and therefore assumed a less invasive, more stable/epithelioid behavior (doi.org/10.3389/fonc.2021.662806, doi.org/10.1016/j.ccell.2023.02.016).

We added a comment to address this point in the results section of Figure 3 of the manuscript :

“Nevertheless, we have encountered some exceptions. In stage II/III some patients relapsed, correlating with their high metastatic potential, whereas others responded very well to chemotherapy, suggesting that although the patient's cells had metastatic potential (possible circulating tumor cells and residual disease), adjuvant chemotherapy effectively mitigated that potential. In Stage IV the discrepancy between staging and metastatic potential could be due to the heterogeneity of the sample that we had access

(clones with low metastatic potential). Another hypothesis is that these tumor cells were already in the process of partial MET (mesenchymal to epithelial transition) rather than EMT (epithelial to mesenchymal transition), and therefore assumed a less invasive, more stable/epithelioid behavior (30,31).”

7. The authors have included the sensitivity threshold of 1.34 in Figure 5 in the assessment of alternative therapies. While this thresholding seems reasonable in the initial studies to define response, justification for continuing to use this bar as a measure for additional or different therapies needs to be provided. This threshold may limit the ability to detect more incremental responses, which may still be of significant clinical value particularly in the context of personalized treatment for high-risk patients.

Thank you for your comment and pointing out this issue. However, in this Figure our goal was to make another point: that although the different treatments have been shown to have similar average response rates in multiple clinical trials (and therefore are considered equivalent choices in international guidelines), individually tumors can respond differently – i.e., can be sensitive to one treatment but resistant to other, just in terms of tumor cell sensitivity.

Nevertheless, if we consider the decision tree model, one patient from this figure (P#41CCU) would be now re-classified (stage III and absence of micrometastases) as no-progression, independent of treatment. Therefore, in the case of hypothetically being treated with FOLFIRI this patient would NOT progress due to absence of metastatic potential. Regarding the graphs, we find it more appropriate to omit the statistical analysis, as it was not employed as a means to assess an effect in our study.

8. Fig 5c/legend: The table only lists zebrafish response (sensitive/resistant) to each treatment. It would be beneficial to include an additional column with the patient response matched to each zAvatar.

Thank you for your comment we have now added a column with the patient response

zAvatar	patient treatment	outcome	option
41CCU	FOLFOX	NO-PRG	FOLFIRI
64CCU	CAPOX	PROG	CAPIRI
203CCU	FOLFIRI	PROG	CAPOX+CET
256CCU	CAPOX	NO-PRG	CAPIRI
294CCU	CAPOX	NO-PRG	FOLFIRI
296CCU	CAPIRI	PROG	CAPOX
61AS	FOLFIRI	PROG	FOLFOX
95AS	FOLFOX	NO-PRG	FOLFIRI
110AS	FOLFIRI	PROG	FOLFOX
225AS	FOLFOX	NO-PRG	FOLFIRI

9. Line 403-417: “In the context of stage II/III, the metastatic potential variable emerges as a critical factor to improve the zAvatar-test accuracy. Here, patients whose zAvatars had no micrometastases are immediately classified as having no-progression disease. This means that their sensitivity to therapy is irrelevant for progression outcome, suggesting that these patients may be spared from chemotherapy, and its toxic side effects.”

While personalization of cancer therapy is desired, as is a reduction in side effects, particularly if the drug is ineffective, the conclusion here is confusing. Those patients with Stage 2/3 disease responded to therapy as predicted by their zAvatar, so why eliminate chemotherapy for this population?

This does not appear a sound recommendation based on the data presented. It would be more helpful to predict for which patients chemotherapy is ineffective, as this

population would be subjected to risk of toxicities without clinical benefit. As such, they should not be given chemotherapy, but rather an alternative treatment be considered.

Thank you for your comment, we agree with your comment and apologize for not being clear.

Some of these stage II/III patient's zAvatars were indeed sensitive to chemotherapy and they did not progress whereas others were resistant but **also did not** progress. The multivariate analysis revealed that in these early-stage patients tumor cells had no metastatic potential and therefore should be classified as no-progression.

Our point was that even in patients that were sensitive to treatment, their outcome would be NO-progression because their tumor cells did not show any metastatic potential. That is why we suggest that "sparing patients from unnecessary chemotherapy when their zAvatars indicate no-metastatic potential, reducing the risk of toxicities of unnecessary treatments".

We will change the text to be clearer to:

"In the context of stage II/III, the metastatic potential variable emerges as a critical factor to improve the zAvatar-test accuracy. Here, patients whose zAvatars had no micrometastases are immediately classified as having no-progression disease. This suggests that in these cases, sensitivity to therapy is irrelevant for progression outcome, suggesting that these patients may be spared from chemotherapy and its toxic side effects."

9b-"It would be more helpful to predict for which patients chemotherapy is ineffective, as this population would be subjected to risk of toxicities without clinical benefit. As such, they should not be given chemotherapy, but rather an alternative treatment be considered. "

We agree with this comment; but besides predicting ineffective chemotherapy, what our data is suggesting is that patients whose tumor cells lack metastatic potential may be spared from toxic treatments.

This kind of rationale is being thoroughly investigated, particularly in the context of rectal cancer. For instance, in rectal cancer management, patients are treated with chemoradiotherapy and those showing a good response may enter a program of "watch&wait" to avoid a debilitating radical surgery.

We might argue that patient without metastatic potential have lower risk of progression and may not require chemotherapy, but it is also worth considering the administration of chemotherapy as a precautionary measure. However, nowadays there are many patients that want to avoid chemotherapy at all costs. In this regard, employing a zAvatar-test would lead to a more confident decision-making.

10. Methods: Graphs display the average of fold changes in Caspase 3 staining – this may not be the optimal way to analyze this data. Moreover, cells are counted manually, which can lead to inaccuracies, particularly when trying to evaluate in 2-D something that exists in 3-D. Authors should comment on these approaches and consider complementing with additional analyses to confirm consistency in their findings.

We appreciate the observation regarding our data analysis methods.

We acknowledge the potential limitations of manual cell counting and have made efforts to explore various automated analysis methods. However, we encountered challenges as none of these automated methods were able to faithfully replicate what the human eye observes. For instance, when distinguishing the nuclei of two overlapping cells, automated methods tend to count them as a single cell, whereas in reality, there are two separate cells.

In many published works, cell death or tumor size is often analyzed using unspecific methods, such as measuring tumor volume from fluorescence intensity. While lipophilic dyes are valuable, their reliability may vary as certain cells stain more effectively than

others, and dead cells or debris may accumulate in the yolk sac, leading to misleading signals. For instance, phagocytic cells like macrophages can become stained after 'consuming' debris, resulting in false-positive signals, including micrometastasis.

Despite the laborious and time-consuming nature of confocal microscopy and manual quantification analysis, we believe that it is important for unequivocal detection of human cells and an accurate interpretation of the results.

Regarding the graphs, the fold change averaging provides a representative measure that summarizes the overall trend in Caspase 3 staining across the entire sample in relation to controls. In our opinion, the fold change represents a standardized way of reporting results as it facilitates comparisons between different experimental conditions.

We have been trying different methods of quantification, including automated quantifications but until now manual counting was the best one with reduced error.

Also, we believe that the ultimate confirmation that our method is robust is the correlation with clinical data: 90% correlation. Nevertheless, as requested we re-analyzed all our data and quantified tumor area of the DeepRED staining and h-Mito (see FigR2) (ROI), and this was not correlating with patient outcome.

Minor:

11. "With exception of some success cases, current cancer molecular and genetic biomarkers have proven insufficient when it comes to reliably predicting treatment outcomes. Most cancer patients do not benefit from genomic precision medicine due to a combination of factors, including the absence of targetable mutations, the lack of effective drugs for specific promising targets and also the possible genetic interactions that may occur between different tumor subclones or with the tumor microenvironment (9, 10)." This comment undersells clinical advances that have been achieved in some cancer types through molecular profiling and moreover the zAvatar system provides an in vivo platform for functional validation of molecularly targeted therapeutics. The language should be changed to better reflect these realities.

Thank you for your comment we will tune down this point. This point is thoroughly discussed by Anthony Letai and colleagues (*Deng et al, 2007*
DOI: 10.1016/j.ccr.2007.07.001; *Letai, 2022* DOI: 10.1158/2159-8290.CD-21-1498), arguing in favor for a combination of molecular profiling together with functional tests. We have now added to the introduction:

"With exception of some success cases, current cancer molecular and genetic biomarkers have proven insufficient when it comes to reliably predicting treatment outcomes. It has been shown that even genetically identical CRC may have differential response to therapy, implying that the basis for therapy response is not only genetic (9). Many cancer patients do not benefit from genomic precision medicine due to a combination of factors, including the absence of targetable mutations, the lack of effective drugs for specific promising targets and also the possible genetic interactions that may occur between different tumor subclones or with the tumor microenvironment (10, 11). Thus, a combination of molecular-profiling precision medicine together with a functional test, where tumor cells are directly challenged with the predicted therapies is fundamental for a more accurate personalized medicine (10, 12)."

12. Results, lines 231-232, Please clarify what is meant by the word "blindly" here. "zAvatar response to treatment was blindly compared with patient clinical response 12 months after starting chemotherapy (Fig. 1a)."

When we refer that the response to zAvatar treatment was "blindly compared" with the patient's clinical response, it means that the comparison was done in a blinded manner. In this context, "blindly" indicates that the experimental researchers (zAvatar Lab) had no previous information about the clinical outcome of any patient at the time of the experiment-the Lab only has info on the type of therapy given to the patient. After the

zAvatar test is performed, the zAvatar-test results were sent to the physicians who previously classified as stable or progression. This assessment by the physicians is done retrospectively after a 12-month follow-up (sometimes more), based in imagiological findings, clinical assessment or/and histological confirmation. Our goal was to reduce any possible biases in the analysis.

We have added a small description in the methods section to clarify:

Response to zAvatar treatment was blindly compared with the patient's clinical response. Experimental researchers had no previous information about the clinical outcome. After performing the zAvatar-test, results were sent to the physicians to analyze correlation.

13. Figure 1D – the cross-sectional CT images of patient#138 are not at the same anatomic level pre- and post-treatment, making correlation regarding response difficult to assess.

Thank you for your comment regarding the figure in question. The initial CT scan image shows the patient's primary tumor at the level of the right colon. After surgery, presenting the right colon becomes impossible since it has been removed. We considered that it would be of little use to present the same section on the CT scan because there is no structure present. Thus, we chose to place a section of the patient's liver image, as the liver is the site most frequently affected by metastatic disease in tumors of the right colon. In this particular case, the liver section shows an absence of signs of disease recurrence, as described in the text. We have added a small text in the legend to clarify this point.

14. Figure 1G' -there is significantly more apoptosis in the control group compared to the treated group? Can the authors explain? Are cells naturally dying?

We can only speculate, but we have seen a few cases where this happens. There could be several reasons why there might be more apoptosis observed in a control or untreated group compared to a treatment group. It is possible that the untreated group experienced natural fluctuations in apoptosis levels due to individual differences among the cells, such as genetic heterogeneity or interactions with the tumor microenvironment.

In addition, the zebrafish host innate immunity may induce apoptosis in the human cancer cells in the absence of a specific treatment. But then treatment inhibits the innate immune system (chemotherapy can lead neutropenia etc) and lead to reduced levels of apoptosis due to impairment of immune response. We have observed a similar phenotype regarding implantation – there are some cases where in the end of the experiment there were more tumors to analyze in the treatment group than in the control (see Póvoa *et al*, *Nat Comm* <https://doi.org/10.1038/s41467-021-21421-y>), we believe that this is due to the innate immune system.

Finally, the treatment itself may have unintended consequences in that particular tumor, potentially activating survival pathways or inducing protective responses in the cancer cells, leading to lower observed apoptosis levels compared to the untreated group.

15. Figure 4, d and i, need to define "implantation rates".

We defined the tumor implantation in the methods section as follows:

$$\% \text{ implantation} = \frac{\text{n}^{\circ}\text{xenografts at 3 dpi with a tumor mass}}{\text{total n}^{\circ}\text{xenografts at 3dpi}} \times 100$$

16. Results, lines 345-346- "This patient presented liver progression three months after completing chemotherapy, matching with the results from the liver metastasis zAvatar-test." language needs to be changed here as it sounds as if there was a 3-month ZF study conducted in parallel.

Thank you for pointing this out. We have revised it to:

“This patient exhibited liver progression three months after completing chemotherapy, matching with the results previously obtained from the zAvatar-test.”

17. Consider moving Results Section 5 “alternative therapies” to the end of the Results section following the Decision Tree analysis to improve the flow of the paper.

Thank you for the suggestion, however we disagree because in Fig5 we did not include the metastasis analysis – our point was just to show the different tumor sensitivities. And we really would like to finish with the tree decision model.

18. Figure 6 legend (line 426) should be (c) instead of (a).

Thank you for pointing this out. We have changed it.

19. Methods: Whole mount immunofluorescence: Are both secondary antibodies anti-rabbit (for Caspase3) and anti-mouse (for h-MITO) Alexa 488 (green)? Was the anti-mouse 647 used for h-MITO when injecting cells into the *fli1a:eGFP*?

In general we use secondary antibodies anti-rabbit 594 to detect Caspase3 (rabbit) and anti-mouse 488 for h-MITO (mouse), even in the *Tg(fli1a:eGFP)* background because it is easy to distinguish the blood vessels from human mitochondria (see Fig. R5).

Figure R5. Example of immunostaining of a zAvatar in *Tg(fli1a:eGFP)*.

Reviewer #2 (Remarks to the Author): with expertise in zebrafish, cancer

We would like to thank reviewer#2 for the critical and careful reading of our manuscript and the opportunity to address all concerns raised, improving our manuscript.

The manuscript from Costa et al., "The zAvatar-test forecasts patient's treatment outcome in colorectal cancer: a clinical study towards personalized medicine," demonstrates the utility of zebrafish patient-derived xenograft models (zAvatar) as a fast predictive platform for personalized treatment in colorectal cancer. The major strength of this research is that it, to date, has performed the most extensive co-clinical study of the link between various data points from patient-derived zebrafish xenograft and overall patient outcomes and showed several properties of the PDZX are statistically linked with patient outcome. The methodology of the study is sound, and the conclusions are well supported by the data. This manuscript is significant in the oncology field, as the zAvatar model may hold the potential to enhance personalized medicine by providing clinicians with additional data sets that can be used to optimize treatment options for each patient.

Overall, the manuscript is well-written and straightforward. Some minor issues should be clarified during the revision process:

1. It will be helpful to state in the methods or results the degree of blinding that is carried out in this study. For example, were researchers who carried out the zAvatar test blinded to the type of patient sample? What about the team that classified progression/no progression disease?

To conduct the test, the researchers in the zebrafish lab were ONLY informed about the type of sample (colon, rectum, or liver metastasis) and the type of treatment administered after sampling (surgery). No team member conducting these experiments had access to the patients' clinical records.

The zAvatar results, along with the anonymized patient identification codes, were later sent to the physicians who classify them as no-progression or progression. This assessment by the physicians is done retrospectively after a 12-month follow-up (sometimes more), based in imagiological findings, clinical assessment and/or histological confirmation.

We have added a small explanation into the methods section:

Response to zAvatar treatment was blindly compared with the patient's clinical response. Experimental researchers had no previous information about the clinical outcome. After performing the zAvatar-test, results were sent to the physicians to analyze correlation.

2. Were the patient cells always injected at the same concentration of cells? If yes, please state how many cells were injected in the methods, and if not, please explain how cell numbers were chosen. The number of injected cells seems inconsistent in the figures. For example, in Figure S2, one larvae has many cells, and another has fewer. Please explain. Would the number of injected cells affect the conclusions drawn about angiogenesis, tumor size, and number of apoptotic cells?

Thank you for your comment. We have been struggling to inject always at the same concentration of cells. However, even with cancer cells lines this is almost impossible due to technical issues, such as needle clotting and gravity induced concentration of the sample in the needle.

To overcome this, we do try to inject all fish with the same volume of cells (we try to inject a volume of cells that matches the zebrafish eye size as a reference), but because of the technical issues referred above we perform an additional quality control check at 1 day post injection (dpi). At 1dpi, we sort xenografts in different sizes and then within each size distribute xenografts to control and untreated groups.

We described all our methods in 2 publications (*Martinez-Lopez et al, 2021* doi: 10.3791/62373; *Costa et al, 2022*, <https://doi.org/10.1002/cpz1.415>).

Nevertheless, at 3dpi (as in these images), the number of cells will inevitably differ because, depending on the tumor cells, some will die, and others will be cleared by the host's immune system, and others will thrive. We also observe this in cancer cell lines, but in patients' samples due to their heterogeneity this is more evident.

“Would the number of injected cells affect the conclusions drawn about angiogenesis, tumor size, and number of apoptotic cells? “

Yes, this could be possible, this is why we added the additional quality control check at 1dpi, to sort xenografts in different sizes and then within each size distribute xenografts to control and untreated groups. Then, we always compare treated xenografts versus matching controls from the same original tumor size-this is why we normalize to the controls- and then present fold change. Although we do this as a precaution, we never saw a major impact of original number of injected cells in angiogenesis, tumor size or apoptosis %, we only observe this “size matters” in studies of innate immune evasion.

3. In the methodology (line 599), the authors say that the tumor size and apoptosis were measured by the number of cells. However, the graphs show tumor size and caspase-3 as fold change. Please clarify how the analysis is done.

We apologize for not being clear. We quantify tumor size by manually counting the number of DAPI/DeepRED cells in the tumor region. Similarly, we assess apoptosis by counting all cells marked with the caspase-3 antibody. Then we divide the number of caspase3 positive cells by the number of tumor cells, obtaining the % of caspase3 in each zAvatar. After calculating the respective percentages in the control and treatment groups, we normalize all percentages by dividing them by the untreated-control mean: a fold change greater than 1 indicates an increase in the treatment group compared to the control; A fold change less than 1 indicates a decrease in the treatment group compared to the control; A fold change of 1 indicates no change in relation to untreated control.

Control-untreated tumor cells may have different baseline apoptosis, with some tumor cells with very low basal apoptosis and others with very high basal apoptosis. Thus, to quantify the impact of treatments in tumor cells we want to compare how much the treatment induced apoptosis in relation to control cells.

For example, one patient 's tumor cells can have 4% basal apoptosis and then upon treatment apoptosis goes up to 8% (this would be 2-fold change) but in another patient, tumor cells may have 30% of basal apoptosis but upon treatment increases to 35% (only 1,16 fold change). So, the first patient although had lower apoptosis had a higher response to treatment. This is why we used fold change and not the absolute % values of apoptosis.

The same rational applied for the tumor size.

4. The authors should clarify why they labeled both tumor cells (DeepRed) and human mitochondria (h-MITO). I would expect that all injected cells would be double-labeled. However, in Figures 2, 4, and 5, there are cells labeled with one stain but not the other. Please explain this.

Thank you for your comment. We will try to clarify to the best of our knowledge and experience.

In our experience, labelling human cancer cells with lipophilic dyes (DeepRed or DiI) can be very heterogeneous, even in cell lines, patient samples can be even more heterogeneous. Some cells take up better some dyes than others. Also, the brightest cells (and that are better visualized in our images) are cells that are dying, to visualize the low intensity staining we would have to overexpose the images.

We use hMITO antibody as a quality control to confirm the presence of human cells, since phagocytes may phagocytose human cells and take up the die and therefore emit a signal that is no longer from human tumor cells but actually zebrafish phagocytes-and therefore can give false-positives. Therefore, we believe that because of these 2 reasons (heterogeneity of the DeepRed signal and higher intensity in dead cells) as well as phagocytosis the 2 signals not always co-localize.

In addition, the h-Mito antibody (MAB1273) is designed to target the surface of intact mitochondria, meaning that its specificity is directed towards antigens present on the outer surface of mitochondria. The expression levels of mitochondrial antigens can vary among different types of cancer cells and also in healthy cells (*Criscuolo et al, 2021, doi: 10.3389/fonc.2021.797265, Chen et al, 2021, https://doi.org/10.1038/s41392-023-01546-w*). In fact, when we started working with this hMITO antibody we were also surprised by the heterogeneity of the hMITO signal and therefore we performed an immunofluorescence of the original patient sample in a paraffin section (**FigR1**)—I hope you can appreciate the heterogeneity of hMito in normal tissue and tumors (see figure below).

Figure R1 – Heterogeneity of hMito staining in paraffin sections of a CRC patient sample

5. From the supplemental table, it seems that only 2 patients received bevacizumab or cetuximab. Yet, the methods state that these antibodies were included in the injection mix for all samples. Please clarify why this was done.

We apologize for not being clear. Inclusion of these antibodies in the injection mix was only performed in cases where patients had undergone treatment with these agents.

We will replace the sentence with:

“Besides the addition to the E3 medium at 1dpi, when patients underwent treatment involving bevacizumab and cetuximab monoclonal antibodies, these agents were also added into the cell suspension prior to injection at of 100 ng/mL and 20 µg/mL, respectively (11, 39).”

6. Authors justify that tumor shrinkage in zAvatars did not predict the absence or presence of disease progression, possibly due to the very fast assay that may not allow sufficient time for effective tumor clearance. Why did the authors choose to treat the zAvatars for just two days? Other papers from the same authors typically use a three-day treatment for the zebrafish xenografts. Please explain this inconsistency.

Patient derived xenografts tend to die more than cell-line-derived xenografts, reducing the number of animals that reach the end of the assay, compromising the zAvatar test. Therefore, we performed several experiments to compare 3 days vs 2 days of treatment and found that the apoptotic response was already statistically significant with only 2 days of treatment (see **FigR6**). This allows us to reach the end of the assay with more zAvatars to analyze.

Although in this example we can already observe a reduction of the tumor size at 3dpi, at 4dpi it is more evident. Nevertheless, we acknowledge that this might not be enough time to observe the impact on tumor size on other cases. In fact, we also observed that for other treatments we have to extend the assay to 6dpi to observe tumor shrinkage although the apoptosis induction was already detected a few days before (see *Figure 6 from Oliveira et al JACS2020 DOI: 10.1021/jacs.0c01622*).

Figure R6. Comparison of treatment response at 3dpi and 4dpi of HCT116 xenografts treated with FOLFIRI. Each dot represents a xenograft.

Finally, shortening the assay allowed us to fit the zAvatar-test into the developmental window of 5dpf (2010/63/EU). All our SOPs (go to 14dpf) were submitted to the regulatory authorities and ethics committees and comply with all standard 3R procedures. Nevertheless, we try to stay within the 5dpf window.

7. Do the authors have any explanation why, in some patients, the increase of activated caspase is correlated with a decrease in tumor size (Figure 5) and in other situations, it is not (Figures 1 and 4)?

Thank you for your comment. We do not know exactly why this happens; we can only speculate. To reduce tumor size, cells need to be cleared by innate immunity, in particular macrophages. It could be that some tumor cells express more “don’t eat me” signals than others that could delay the clearance process.

8. In Figure 4, the authors also concluded that both samples of P#229CCU were sensitive to FOLFOX treatment. However, there is no statistical analysis in graph B. In the second case (P#189AS), authors concluded that zAvatars derived from the primary tumor were sensitive to FUFOL treatment, and those derived from liver metastasis displayed resistance. Again, there is no statistical analysis in graph G. Please add the statistics used to draw these conclusions.

Thank you for your question. We did not include statistical analyses, as we determined sensitivity/resistance based on the apoptosis fold change threshold of 1.34, as explained previously in Figure 2. In other words, even if the statistical analysis is non-significant (which can sometimes be due to the low sample size), if the threshold is above 1.34, zAvatar is considered sensitive (depicted by the red line in the graphs). However, acknowledging the significance of statistical information in data interpretation, we will incorporate this information into the figure legends.

9. In Figure 7 and the text, it is unclear if the authors used the multivariate analysis threshold to classify sensitive and resistant avatars.

Thank you so much for this observation. This analysis was previously carried out based on the apoptosis threshold of 1.34 but also according to the decision tree, but by mistake the final figure had the previous graphs based on 1.34. Thanks so much for spotting this! We have now updated this figure with the correct data taking into account the classification of the decision tree model.

Figure 7. Patients with a sensitive zAvatar-test have longer Progression-Free Survival. (a) Kaplan-Meier survival curves comparing the PFS of patients based on sensitivity or resistance of their zAvatar-test (taking into account the tree decision model). The PFS was calculated from the initiation of chemotherapy until either last observation or date of progression. (b) When analyzing patients from all stages, the zAvatar sensitive group had a longer mean PFS of 30.9 months compared to 7.5 months for the resistant group (n=55; $p < 0.0001$). (c, d) Similarly, in stage II/III patients the mean PFS was 37.0 months versus 11.3 months (n=32; $p < 0.0001$), and in stage IV patients the mean PFS was 11.4 months versus 5.9 months (n=23; $p = 0.0063$).

10. Overall, some of the text in figures is very small and difficult to see even on zoom, particularly in Fig 2D-E. Other figures are pixelated, such as the graphical abstract and graphs next to images. These should be fixed if possible.

We apologize for this; we have now increased the text and also fixed the quality of the images – that were compressed in the submission process.

Reviewer #3 (Remarks to the Author): with expertise in colorectal cancer, therapy

We would like to thank reviewer#3 for the critical and careful reading of our manuscript and the opportunity to address all concerns raised, improving our manuscript.

1. Enrolled patients included Stage II/III and Stage IV patients, there are many issues arising from this decision

a. Goals of therapy are different for Stage II/III versus Stage IV patients. The endpoint in adjuvant therapy is usually disease recurrence and is usually reported as 3 year or 5 year recurrence free survival. It is not clear whether “no evidence of disease recurrence within 12 months after treatment initiation” can be used as evidence of “no progression”.

Thank you for your comment and we recognize the complexities of having both Stage II/III and Stage IV patients in our study.

We acknowledge that the goals of therapy differ between these stages. Thus, when referring to Stages II/III, we consider adjuvant therapy, whereas for Stage IV, we specifically mention postoperative therapy. This distinction accounts for the different goals and contexts associated with each stage. We also believe that, although the goal of the treatment is different, our ability to understand whether the model reveal the sensitivity to the treatment will be little affected even when we perform treatments with different goals.

In our study, the criterion 'no evidence of disease recurrence within 12 months after treatment initiation' was used to assess short-term progression outcomes. This criterion of 12 months was established because we thought that it would be too ambitious to expect that a functional test would be able to predict clinical outcome more than 1 year. Our zAvatar-test is a very short assay - like a snap-shot of the tumor at the time-point of sample collection where we assess tumor sensitivity and metastatic potential – but our assay does not allow for tumor evolution and therefore we did not expect to have long-term predictive value.

We recognize that the term 'no progression' might not be correct in clinical practice with this specific follow-up duration. However, we needed a way to distinguish between responders and non-responders patients. Initially, we used the term 'stable,' but following advice from physicians in our team, we adjusted to 'no-progression' to better align with established terminology.

We propose to change in the text:

“no evidence of disease recurrence within 12 months after treatment initiation” can be used as evidence of “no-progression within 12 months(NO-PRG)”.

b. 6 of 32 Stage II/III patients had recurrence within 12 months. This number seems to be high, compared to those reported in the literature.

Thank you for your comment. These 6 patients were Stage III, none of the Stage II recurred in 12 months.

The totality of patients who are classified in stage II and III is a very heterogeneous group and therefore has very different rates of disease-free survival or relapse. These differences are reflected in the literature, depending on the population presented. As an example, patients staged as T4bN2a, therefore stage IIIC, have recurrence rates greater than 70% (*Gunderson et al, 2010, doi: 10.1200/JCO.2009.23.9194*), however patients staged as T1N1a have recurrence rates of less than 10% (*Gunderson et al, 2010, doi: 10.1200/JCO.2009.24.0952*).

In general, when we analyse at the stage III group as a whole, a recurrence rate of the disease, even in the first year, of about 18% (6/32), as presented in the study, seems to be within what is expected for the prognostic group presented.

For example, a recent study by Nors J et al, JAMA Onc 2024 (doi: 10.1001/jamaoncol.2023.5098) reported that “For colon cancer, the 5-year CIF of recurrence decreased over the 3 calendar periods from 16.3% to 6.8% for UICC stage I, from 21.9% to 11.6% for UICC stage II, and from 35.3% to 24.6% for UICC stage III colon cancer.” Thus 24,6% for Stage III is not so different the 18% that we observed in our smaller cohort.

2. Pre-treatment and post-treatment CT images were from different areas. For Patient 138CCU, the patient underwent surgical resection for colon cancer. Pre-treatment and post-treatment images were not helpful.

Thank you for your comment regarding the figure in question. Our goal was to show/illustrate that there was no progression. The initial CT scan image shows the patient's primary tumor at the level of the right colon. After surgery, presenting the right colon becomes impossible since it has been removed. We considered that it would be of little use to present the same section on the CT scan because there is no structure present. Thus, we chose to place a section of the patient's liver image, as the liver is the site most frequently affected by metastatic disease in tumors of the right colon. In this particular case, the liver section shows an absence of signs of disease recurrence, as described in the text.

We have added a small text in the legend to clarify this point.

3. Figure 1 E shows n =8, while Figure 1 G shows n = 19, it is not clear why the number of repeats is different. This seems to be the pattern throughout the manuscript, experiments in different patients/different chemo were performed different times.

We apologize for not being clear.

N=8 and N=19 depict the number of zAvatars that reached the end of the assay and were quantified. The original number of xenografts that is possible to generate depends on the amount and quality of the original tumor sample. There are some patient's samples that allows the generations of dozens to hundreds of zAvatars and others much fewer.

We changed the legend of Figure 1 and also Figure 4 and 5 to be clearer:

“Each dot represents one zAvatar and the total number (N) of zAvatars analyzed is indicated in the images”

4. Based on Figure 1, there appears to be significant intra-subject (zebrafish) variation in apoptosis FC. Figure 2D and 2E seemed to be generated from average values. Indeed, Figures 2D and 2E were generated from average values. Each zAvatar from a patient is represented by a fold change value of apoptosis, indicating the apoptosis level induced by the treatment. This value is obtained by averaging the apoptosis values across all zAvatars corresponding to a specific patient."

Thank you for your comment. Yes, we acknowledge this variability, we believe that the variability of our data is capturing the inherent tumor heterogeneity that is reflected in different tumor behaviors and responses to therapy.

Moreover, it's important to note that we inject not only tumor cells but also their tumor microenvironment (TME). This complexity reflects the diverse nature of patient responses, and despite efforts to standardize conditions, individual variations are always expected taking into account tumor biology. Nevertheless, our data shows that taking into account the AVG response is highly correlative of clinical response, therefore we are confident on the robustness of our data.

5. Similarly, Figure 4 B/C/G/H show large intra-subject variability. For example, there appears to be an outlier in Figure 4B, rectum/FOLFOX group. Differences between control and FOLFOX were likely driven by this outlier.

Thank you for your point. We acknowledge the presence of outliers and we thoroughly examined our data by using the tool “graphpad outlier” to identify these possible outliers and removed them. So, the “apparent outlier”, although it may look as an outlier it was not according to the statistical tool that we used.

We have now included in the Methods:

Whenever a value suggestively deviating from the dataset's mean was observed, a comprehensive examination of the dataset was conducted using the "GraphPad Outlier" tool (<https://www.graphpad.com/quickcalcs/Grubbs1.cfm>).

6. 79 patients were enrolled in the study, zAvatar-tests were successful in only 55 patients. Results presented in Figure 2F and 2G only included 55 patients.

The results presented in Figure 2F and 2G are based on this subset of 55 patients due to successful zAvatar-test completion. Indeed, despite recruiting 79 patients, we were able to conduct the test only in 55 patients. This was attributed to various factors, some patient samples had high levels of necrosis impairing xenograft generation, zAvatar mortality during the assay, or low implantation rates.

We have now included the explanation in the beginning of Results section:

“The main reasons for nonsuccess were a small initial tumor sample, sample necrosis or death of zAvatars during the experiment. Patients whose zAvatars had low implantation ($n < 4$ zAvatars for each condition) were excluded from the study.”

7. Since Stage II/III patients have different prognosis compared to Stage IV patients, it is not surprising that tumor stage is identified as a factor in multivariate analysis.

Thank you for your comment, we also agree, but we are only showing what the multivariate analysis revealed. Nevertheless, although evident, without the unbiased multivariate analysis we did not realize that we had to divide patients in the 2 different stages, and they would have different apoptosis thresholds or that metastatic potential would be so revealing in early-stage patients.

Interestingly, other factors that might seem obvious, such as tumor resection margins (R0/R1), did not emerge as statistically significant predictive variables.

REVIEWER COMMENTS

Reviewer #1 (Remarks to the Author):

The authors have done a reasonable job responding to many of the comments raised in the prior reviews. However, there remain a few outstanding issues to be considered.

1. Tumor shrinkage does not predict response, and the authors state that the brightest cells are those that are dying and are then phagocytosed. What experimental evidence do they have to support this contention? Can they show double staining of these brightly labeled cells with Caspase 3 or inject into a macrophage reporter line to show phagocytosis of this cell population?
2. It would be helpful to include timelapse imaging to demonstrate that distant cells represent true metastases rather than cells that migrated from a PVS injection since the authors acknowledge that in some cases cells are present in circulation following injection but tend to disappear. This timelapse would complement the co-labeling experiment suggested above.
3. The conclusion "In the context of stage II/III, the metastatic potential variable emerges as a critical factor to improve the zAvatar-test accuracy. Here, patients whose zAvatars had no micrometastases are immediately classified as having no-progression disease. This suggests that, in such cases, sensitivity to therapy is irrelevant for progression outcome, suggesting that these patients may be spared from chemotherapy and its toxic side effects" remains insufficiently supported by the data presented, certainly not to the degree that a definitive clinical recommendation should be made. This language needs to be revised as at least a subset of these patients' lack of progression was on account of their response to chemotherapy. Again, it seems like this algorithm would better predict which Stage 2/3 patients would respond to chemotherapy vs. those that would be resistant and progress and might need different therapy.
- 4 Figure 1e - While it is appreciated the patient is post-right hemicolectomy - pre and post treatment CT imaging should show the same anatomic level/region or otherwise are not evaluable or contributory as a comparison. The images should be changed or the panels removed and findings described in the text.

Reviewer #4 (Remarks to the Author):

Thank you for the opportunity to review this excellent and exciting manuscript. Here are some comments.

1: I am a clinician and as such leave the preclinical critique aside. I have read the previous review response and am satisfied with the current revision.

2: The main thing missing here is context. There is only one sentence referencing organoids. Time and costs are mentioned, but no concrete data or results. I would be interested in actually knowing the time and cost differences. Is zAvatar testing cheaper? My understanding is the organoids can be tested within 7-14 days. this is quite similar. Please explain the advantages and disadvantages more succinctly.

3: There is also another method recently published 2023. (Cashin PH, Söderström M, Blom K, Artursson S, Andersson C, Larsson R, Nygren P. Ex vivo assessment of chemotherapy sensitivity of colorectal cancer peritoneal metastases. *Br J Surg.* 2023 Aug 11;110(9):1080-1083. doi: 10.1093/bjs/znad066.). It is exactly in the same field of colorectal cancer chemotherapy resistens. Consider reviewing the differences between these two methods.

4: Considering that progression free survival appears to be quite well associated with differnt chemotherapy sensitivity tests. One wonders about overall survival. Do you have data on this? It could be interesting to note if it the matters. The reason this is important is that "lost time" giving resistant chemotherapy has not really been proven. In other words, the zAvatar test may just tell us what a 3 month CT scan will anyways tell us. The patient will switch to another chemotherapy and then have a response. Rolling through several chemotherapy lines as compared to starting with a sensitive line and then rolling through "resistance" lines may still result in the same overall survival in the end. This is the problem with only using progression-free survival. In the end, all patients who tolerate chemotherapy well end up anyway testing the relevant chemotherapy lines regardless of initial sensitivity. It would be interesting to see if a sensitivity test and the use of the "best" treatment first actually results in a changed overall survival outcome as well.

Dear Reviewers,

We would like to thank reviewers for their time and critical reading of our manuscript and the opportunity to address all raised concerns, improving our manuscript.

Reviewer #1 (Remarks to the Author)

Dear Reviewer#1,

We would like to thank you once again for your time and critical reading of our revised manuscript and the opportunity to address all raised concerns, which allowed us to improve our manuscript.

The authors have done a reasonable job responding to many of the comments raised in the prior reviews. However, there remain a few outstanding issues to be considered.

- 1. Tumor shrinkage does not predict response, and the authors state that the brightest cells are those that are dying and are then phagocytosed. What experimental evidence do they have to support this contention? Can they show double staining of these brightly labeled cells with Caspase 3 or inject into a macrophage reporter line to show phagocytosis of this cell population?

Thank you for your comment. We have selected two patient examples to illustrate this further. In **Figure R1a**, tumor cells were injected into zebrafish transgenic line *mpeg1:mCherry*, a macrophage reporter line. I hope you can appreciate that the majority of the brightest cells labeled with Deep Red are co-localizing with the macrophages, suggesting that these tumor cells are being phagocytosed by macrophages. Similarly, in **Figure R1b**, the brightest cells are co-localizing with the activated caspase 3 antibody.

Figure R1: (a) Example of tumor cells from P#110 (labelled with DeepRed) injected into the zebrafish transgenic line *Tg(mpeg1:mCherry)*, a reporter for zebrafish macrophages. Yellow arrows indicate the brightest cells, likely undergoing phagocytosis. Blue arrows highlight cells with less intense staining. Host macrophages surrounding the tumor can be observed at the bottom of the image. (b) Example of tumor cells from P#330 (labelled with DeepRed) stained with activated caspase 3 (in green). Yellow arrows indicate the brightest cells, which coincide with cells that are undergoing apoptosis.

2. It would be helpful to include timelapse imaging to demonstrate that distant cells represent true metastases rather than cells that migrated from a PVS injection since the authors acknowledge that in some cases cells are present in circulation following injection but tend to disappear. This timelapse would complement the co-labeling experiment suggested above.

Thank you for your comment. We would just like to clarify that during injection some cells might enter circulation despite being injected in the PVS. Nonetheless, these cells still have to survive the sheer stress, evade innate immune surveillance and then have to extravasate to seed the target site. Thus, we believe that these cells that reach the tail and are still present at 3dpi are cells that indeed have several features that allows them to form micrometastases, and this is what we mean by metastatic potential.

Nevertheless, to address your concern we conducted a new experiment where we injected a liver metastasis sample of a colon cancer patient (P#275) into *mpeg1:mCherry* transgenic zebrafish and recorded a time-lapse movie overnight, focusing specifically on the tail region (**video R1**).

Cells are labeled with DeepRed (purple in the movie) and macrophages are depicted in green (Tg (*mpeg1:mCherry*) *false colour*). In the video, we can observe tumor cells persisting in the tail despite their interaction with macrophages, and other cells that are phagocytosed and subsequently cleared. Thus, within this tumor sample, there are cells with metastatic potential and others that cannot evade the innate immune system and thus are unable to seed distant sites.

3. The conclusion "In the context of stage II/III, the metastatic potential variable emerges as a critical factor to improve the zAvatar-test accuracy. Here, patients whose zAvatars had no micrometastases are immediately classified as having no-progression disease. This suggests that, in such cases, sensitivity to therapy is irrelevant for progression outcome, suggesting that these patients may be spared from chemotherapy and its toxic side effects" remains insufficiently supported by the data presented, certainly not to the degree that a definitive clinical recommendation should be made.

This language needs to be revised as at least a subset of these patients' lack of progression was on account of their response to chemotherapy. Again, it seems like this algorithm would better predict which Stage 2/3 patients would respond to chemotherapy vs. those that would be resistant and progress and might need different therapy.

Thank you for your comment. We acknowledge the need for caution in interpreting our conclusions. Our intention was just to reflect what the data **suggests**, not that clinical recommendations should change. In order to avoid potential misunderstandings, we have removed that sentence.

"In the context of stage II/III, the metastatic potential variable emerges as a critical factor to improve the zAvatar-test accuracy. Here, patients whose zAvatars had no micrometastases are immediately classified as having no-progression disease."

4. Figure 1e - While it is appreciated the patient is post-right hemicolectomy - pre and post treatment CT imaging should show the same anatomic level/region or otherwise are not evaluable or contributory as a comparison. The images should be changed or the panels removed and findings described in the text.

We have now changed the CT scan accordingly.

Reviewer #4 (Remarks to the Author):

Dear Reviewer#4,

We would like to thank you for your time and critical reading of our manuscript and the opportunity to address all raised concerns, improving our manuscript.

Thank you for the opportunity to review this excellent and exciting manuscript. Here are some comments.

1: I am a clinician and as such leave the preclinical critique aside. I have read the previous review response and am satisfied with the current revision.

Thank you for your understanding and consideration.

2: The main thing missing here is context. There is only one sentence referencing organoids. Time and costs are mentioned, but no concrete data or results. I would be interested in actually knowing the time and cost differences. Is zAvatar testing cheaper? My understanding is the organoids can be tested within 7-14 days. this is quite similar. Please explain the advantages and disadvantages more succinctly.

Thank you for your comment.

While both zAvatar testing and organoid testing offer valuable insights for personalized cancer treatment, we would like to highlight some relevant differences:

- Organoid models still lack many complex interactions observed in the tumor microenvironment (TME) or in a living organism. In zAvatars, tumor cells are injected together with the whole TME (we do not select tumor cells), providing a more physiologically relevant TME compared to in vitro models
- Organoids rely on Matrigel, but its use presents several challenges including batch-to-batch variability (Kozlowski et al, doi.org/10.1038/s42003-021-02910-8; Zhao et al doi.org/10.1038/s43586-022-00174-y; Kim et al, doi.org/10.1038/s41467-022-29279-4).
- Zebrafish embryos are transparent, allowing real-time imaging of tumor growth, invasion, metastases formation and response to treatment.
- According to the literature organoid testing typically requires 4-5 weeks in average (Pollock et al, DOI: ; Maenhoudt et al, DOI: 10.1016/j.xpro.2021.100429), with some rare exceptions yielding results within 7-14 days (Wang et al, 2023; Pleguezuelos-Manzano et al, doi.org/10.1002/cpim.106). Regarding costs, the creation and maintenance of organoid cultures can be resource-intensive: the culture media is very rich in expensive growth factors and Matrigel, contributing to higher overall costs (Rezakhani et al, doi.org/10.1016/j.biomaterials.2021.121020). According to our calculations and experience organoids cost ~20% more than zAvatars but can take much more time to develop and therefore more challenging to obtain results in a clinically relevant window.
- zAvatars allow to screen for therapies that can only be catalyzed in-vivo, like cyclophosphamide (Steinbrecht et al, DOI: 10.1186/s13568-020-01064-w) and methotrexate (Bedoui et al, DOI: 10.3390/ijms20205023). For these therapeutic agents, in vitro models cannot be used.
- zAvatars allow **evaluation of the metastatic potential**, which cannot be evaluated in organoids.
- Organoids can be cultured long-term, enabling longitudinal studies of tumor evolution and response to therapy, this is not possible in zAvatars, unless using immune compromised hosts.

We added some context on this in the Discussion:

“Fresh sample availability and heterogeneity are probably the main limitations in establishing zAvatars, which are common to all patient-derived models. Mouse and organoid Avatars have shown very similar predictive values (32-38). However, practical constraints such as time, costs, and the use of Matrigel are associated with these models. Additionally, other emerging models such as 3D spheroids (Pasch et al 2019, Miyoshi et al 2018;) and ex-vivo explants (Martin et al 2019; da Mata et al 2021; Cashin et al 2023) have also demonstrated very promising results. Nevertheless, these models collectively lack the complexity of an in vivo system necessary for tracking metastatic potential or screening therapies requiring in vivo metabolism, for instance.”

3: There is also another method recently published 2023. (Cashin PH, Söderström M, Blom K, Artursson S, Andersson C, Larsson R, Nygren P. Ex vivo assessment of chemotherapy sensitivity of colorectal cancer peritoneal metastases. *Br J Surg.* 2023 Aug 11;110(9):1080-1083. Doi: 10.1093/bjs/znad066.). It is exactly in the same field of colorectal cancer chemotherapy resistens. Consider reviewing the differences between these two methods.

Thank you for your comment. We have now included this reference in the Discussion. The ex-vivo assessment of drug sensitivity using the FMCA (Fluorescent Microculture Cytotoxicity Assay) offers several advantages, including the preservation of tissue architecture (although not shown in this publication) and a relatively short turnaround time for obtaining results.

In this study, the authors observed a longer PFS in patients with a drug scored as sensitive than those classified as resistant. However, OS did not differ significantly between the two groups.

To our understanding, one of the significant distinctions between our study and this one is that they do not compare head to head their assay using the same drug as the patient, and therefore they could not calculate the positive /negative predictive value of the assay. As a result, the concordance between ex-vivo sensitivity and in vivo treatment response rates remains to be fully validated.

Moreover, the main difference lies in the fact that the zAvatar assay is conducted in a living organism, encompassing functional organs such as a beating heart, blood and lymphatic systems, liver, bone marrow, kidneys, and central nervous system. In this dynamic environment, tumors can engage in both local and systemic cell-cell interactions. These interactions occur between the tumor and the host, allowing long-distance communication. Consequently, zAvatar allows for the recapitulation of cancer hallmarks such as cell migration, invasion, metastasis, angiogenesis, and immune evasion, which are not observable in vitro settings.

Regarding drug testing, ex-vivo explants enable direct exposure to drugs in vitro, facilitating the assessment of drug sensitivity and resistance within a controlled laboratory setting.

On the other hand, zAvatars offer a more physiologic environment for drug testing because we cannot use very high doses, otherwise would kill the host. Even though drug pharmacodynamics in zebrafish may differ from mammals, many compounds have been shown to block disease in a similar way. This has led to an increasing number of compounds that were discovered in zebrafish screens and now are entering into human clinical trials.

Finally, zAvatars also allow to screen for therapies that can only be catalyzed in-vivo, like cyclophosphamide and methotrexate (Steinbrecht et al, DOI: 10.1186/s13568-020-01064-w; Bedoui et al, DOI: 10.3390/ijms20205023). For these therapeutic agents, organoids, or ex-vivo cultures cannot be used as predictors of treatment response, limiting their use for personalized therapy.

4: Considering that progression free survival appears to be quite well associated with different chemotherapy sensitivity tests. One wonders about overall survival. Do you have data on this? It could be interesting to note if it the matters.

The reason this is important is that "lost time" giving resistant chemotherapy has not really been proven. In other words, the zAvatar test may just tell us what a 3 month CT scan will anyways tell us. The patient will switch to another chemotherapy and then have a response. Rolling through several chemotherapy lines as compared to starting with a sensitive line and then rolling through "resistance" lines may still result in the same overall survival in the end. This is the problem with only using progression-free survival. In the end, all patients who tolerate chemotherapy well end up anyway testing the relevant chemotherapy lines regardless of initial sensitivity. It would be interesting to see if a sensitivity test and the use of the "best" treatment first actually results in a changed overall survival outcome as well.

Thank you for your insightful comment. Indeed, while PFS provides valuable insights into treatment response, the ultimate aim is to improve patient overall survival outcomes and quality of life from a practical clinical point of view.

Our primary goal here was to test the predictive value of the zAvatar-test in forecasting patient clinical outcomes. We specifically evaluated the sensitivity to the initial post-surgery regimen and accessed clinical outcome 12 months after treatment.

We agree that the impact of "lost time" remains unproven in terms of OS. Nevertheless, if a patient is treated with an ineffective therapy that is accessed 3 months later and then changes, it is reasonable to assume that this patient was exposed during 3 months to unnecessary treatments/side effects and incurred unnecessary healthcare costs, potentially resulting in lost therapeutical time and quality of life. However, whether this lost time is relevant in terms of OS is a valid question that warrants further investigation, and we thank you for raising this question.

We did not originally perform the OS analysis because our study was not designed for such examination. Our cohort is small for an OS analysis, we have a relatively short follow up period and may be affected by other confounding variables. Nevertheless, we have now analyzed our data on OS (**Fig. R2**), but please take in account that this might not be correct due to the above-mentioned reasons.

Similarly to PFS, among patients across all stages, those identified as sensitive by the zAvatar-test had a significantly longer OS compared to those classified as resistant (n=55; p=0.0002). Similarly, in stage II/III patients, the sensitive group exhibited a prolonged OS (n=32; p=0.045). However, statistical significance was not observed in stage IV patients (n=23; p=0.1475); possibly due to the small sample size.

Nevertheless, in stage IV we observed that only 2 out of 8 (25%) patients with a sensitive zAvatar-test died, compared with 9 out of 15 (60%) patients with a resistant test. But, again, these numbers are small, and the follow-up period is also very variable and therefore we probably should be very careful in taking any conclusions from these data.

Figure R2. Overall survival of patients with sensitive vs resistant zAvatar-test. (a) Kaplan-Meier survival curves comparing the overall survival (OS) of patients based on sensitivity or resistance of their zAvatar-test. The OS was calculated from the initiation of chemotherapy until either last observation or date of death. (b) When analyzing patients from all stages, the zAvatar sensitive group exhibited a longer OS compared to the resistant group (n=55; p=0.0002). (c) Similarly, among stage II/III patients, the OS was prolonged for the sensitive group (n=32; p=0.045). (d) In stage IV patients, no statistically significant differences in OS were observed (n=23; p=0.1475).

Retrospective real-life study:

To investigate whether the lack of statistical difference in stage IV patients could be attributed to sample size and to analyze if a prolonged PFS correlates with a prolonged OS, we reasoned that we could perform a **small retrospective real-life study** to test this hypothesis. This study did not consider the zAvatar-test but instead relied only patient data.

Here, we selected stage IV CRC patients that underwent post-operative systemic treatment and categorized them into two groups: responders, who did not show disease progression 12m after chemotherapy vs non-responders, who did not obtain clinical benefit from therapy and progressed. The classification of response was conducted by each patient's oncologist and the time-point for assessing response was the same used previously in the for the zAvatar-test: 12 months after treatment. We then analyzed their overall survival (OS) to further elucidate the observed trends (**Figure R3**).

Our findings revealed that the responders group (NO-Progression) exhibited a prolonged OS compared to the non-responders group (Progression) (p=0.0147). This analysis suggests that optimizing therapy choice ("giving the right treatment to the right patient"), could significantly increase OS, even in advanced stages of the disease. However, as before, these numbers are small and the follow-up period is limited, which limits the reliability of any potential conclusions.

Figure R3. Retrospective real-life study. Overall survival (OS) of stage IV CRC patients based on response to post-operative systemic chemotherapy. Blue line: patients with no-progression. Red line: patients who progressed 12 months after treatment ($p=0.0147$, $n=14$ vs. $n=20$, respectively).

In summary, further research and analysis are warranted to clarify the implications of using sensitivity testing into treatment decision-making and its impact on OS. However, our preliminary data results suggest that utilizing a test with a high sensitivity and specificity may indeed influence overall survival outcomes.

Nevertheless, a controlled randomized clinical trial (which we will be launching this year) where oncologist choice is compared with the zAvatar-test-choice should be the ideal setting for this clarification regarding OS.

REVIEWERS' COMMENTS

Reviewer #1 (Remarks to the Author):

The authors have done an admirable job responding to the remaining concerns raised in the review of the last iteration of the manuscript.

1. They have presented additional experimental data demonstrating that the brightest human cells observed in the xenografts are, in fact, dying cells. This data is included in the Response to Reviewers but should be added to the manuscript as a supplemental figure.
2. The time lapse imaging that has been added nicely illustrates the interaction of human cancer cells with zebrafish macrophages. This video should also be included as supplemental data. However this alone does not confirm these cells are metastatic, particularly as a subset of these cells may be directly introduced into the circulation at the time of injection. Thus, it is suggested that the language in Section 3 of the Results referencing micrometastasis is replaced with "micrometastatic potential".

Reviewer #4 (Remarks to the Author):

I have read all the new responses and am impressed with the review work performed. Results are consistent and important issues have been addressed. I have no further comments.

Dear Reviewer#1,

We would like to thank you once again for your time and critical reading of our revised manuscript and the opportunity to address all raised concerns, improving our manuscript.

Reviewer #1 (Remarks to the Author)

The authors have done an admirable job responding to the remaining concerns raised in the review of the last iteration of the manuscript.

1. They have presented additional experimental data demonstrating that the brightest human cells observed in the xenografts are, in fact, dying cells. This data is included in the Response to Reviewers but should be added to the manuscript as a supplemental figure.

Thank you for your comment. This Figure has now been included as Supplementary Figure 2.

2. The time lapse imaging that has been added nicely illustrates the interaction of human cancer cells with zebrafish macrophages. This video should also be included as supplemental data.

Thank you for your comment. The video has been included as Supplementary Movie 1.

However this alone does not confirm these cells are metastatic, particularly as a subset of these cells may be directly introduced into the circulation at the time of injection. Thus, it is suggested that the language in Section 3 of the Results referencing micrometastasis is replaced with "micrometastatic potential".

We have updated the title of section 3 of Results as well as Figure 3 to:
"Metastatic potential in zAvatars correlates with tumor staging and patient clinical progression."